# BICC1 interacts with PKD1 and PKD2 to drive cystogenesis in ADPKD

Uyen Tran[1†], Andrew J Streets[2†], Devon Smith[2†], Eva Decker[3],
Annemarie Kirschfink[4], Lahoucine Izem[1], Jessie M Hassey[1], Briana Rutland[1],
Manoj K Valluru[2], Jan Hinrich Bräsen[5], Elisabeth Ott[6], Daniel Epting[6],
Tobias Eisenberger[3], Albert CM Ong[2]*, Carsten Bergmann[3,6]*, Oliver Wessely[1]*

[1]Department of Heart, Blood & Kidney Research, Cleveland Clinic Research,
Cleveland Clinic, Cleveland, United States; [2]Kidney Genetics Group, Division of
Clinical Medicine, School of Medicine and Population Health, University of Sheffield,
Sheffield, United Kingdom; [3]Medizinische Genetik Mainz, Limbach Genetics, Mainz,
Germany; [4]Department of Human Genetics, RWTH University, Aachen, Germany;
[5]Institute of Pathology, Medizinische Hochschule Hannover, Hannover, Germany;
[6]Department of Medicine IV, Faculty of Medicine, Medical Center-University of
Freiburg, Freiburg, Germany

**\*For correspondence:**
a.ong@sheffield.ac.uk (ACMO);
carsten.bergmann@medgen-mainz.de (CB);
wesselo@ccf.org (OW)

[†]These authors contributed
equally to this work.

## eLife Assessment

This study presents **valuable** findings regarding the basic molecular pathways leading to the cystogenesis of Autosomal Dominant Polycystic Kidney Disease, suggesting BICC1 functions as both a minor causative gene for PKD and a modifier of PKD severity. **Solid** data were supplied to show the functional and structural interactions between BICC1, PC1 and PC2, respectively. The characterization of such interactions remains to be developed further, which renders the specific relevance of these findings for the etiology of relevant diseases unclear.

**Abstract** Autosomal-dominant polycystic kidney disease (ADPKD) is primarily of adult-onset and caused by pathogenic variants in *PKD1* or *PKD2*. Yet, disease expression is highly variable and includes very early-onset PKD presentations in utero or infancy. In animal models, the RNA-binding molecule Bicc1 has been shown to play a crucial role in the pathogenesis of PKD. To study the interaction between BICC1, PKD1, and PKD2, we combined biochemical approaches, knockout studies in mice and *Xenopus,* genetic engineered human kidney cells carrying *BICC1* variants, as well as genetic studies in a large ADPKD cohort. We first demonstrated that BICC1 physically binds to the proteins Polycystin-1 and -2 encoded by *PKD1* and *PKD2* via distinct protein domains. Furthermore, PKD was aggravated in loss-of-function studies in *Xenopus* and mouse models, resulting in more severe disease when *Bicc1* was depleted in conjunction with *Pkd1 or Pkd2*. Finally, in a large human patient cohort, we identified a sibling pair with a homozygous *BICC1* variant and patients with very early onset PKD (VEO-PKD) that exhibited compound heterozygosity of *BICC1* in conjunction with *PKD1 and PKD2* variants. Genome editing demonstrated that these *BICC1* variants were hypomorphic in nature and impacted disease-relevant signaling pathways. These findings support the hypothesis that BICC1 cooperates functionally with PKD1 and PKD2, and that *BICC1* variants may aggravate PKD severity, highlighting RNA metabolism as an important new concept for disease modification in ADPKD.

## Introduction

Autosomal-dominant polycystic kidney disease (ADPKD) is the most frequent life-threatening genetic disease and one of the most common Mendelian human disorders with an estimated prevalence of 1/400–1000 (*Harris and Torres, 2009*; *Ong et al., 2015*). This equates to around 12.5 million affected individuals worldwide. About 5–10% of all patients requiring renal replacement therapy are affected by ADPKD. The majority of ADPKD patients carry a pathogenic germline variant in the *PKD1* or *PKD2* gene and present with the disease in adulthood (*Ong et al., 2015*; *Torres et al., 2007*; *Bergmann et al., 2018*). However, occasionally, ADPKD can manifest in infancy or early childhood (<2 years, very-early onset ADPKD [VEO-ADPKD]), and in late childhood or early teenage years (2–16 years, early-onset ADPKD [EO-ADPKD]) (*Bergmann and Zerres, 2007*; *Ogborn, 1994*). VEO patients and fetuses often present with a Potter sequence and significant peri- or neonatal demise, which can be clinically indistinguishable from a typical autosomal-recessive polycystic kidney disease (ARPKD) presentation caused by *PKHD1* mutations (*Rossetti et al., 2009*; *Vujic et al., 2010*). However, in contrast to VEO/EO-ADPKD, ARPKD kidneys invariably manifest as fusiform dilations of renal collecting ducts and distal tubules that usually remain in contact with the urinary system (*Bergmann et al., 2018*). Co-inheritance of an inactivating *PKD1* or *PKD2* mutation with an incompletely penetrant minor PKD allele in trans provides a likely explanation for VEO-ADPKD (*Bergmann, 2015*). In fact, we recently reported that the majority (70%) of VEO-ADPKD cases in an international diagnostic cohort had biallelic *PKD1* variants (i.e., a pathogenic variant in trans with a hypomorphic, low penetrance variant), while cases of biallelic *PKD2* and digenic *PKD1/PKD2* were rather rare (*Durkie et al., 2021*) in line with the dosage theory for PKD (*Ong and Harris, 2015*). Several other genes, including *GANAB, DNAJB11, ALG8, ALG9,* and *IFT140,* have been associated with a dominant, but late-onset atypical adult presentation and sometimes incomplete penetrance (*Bergmann et al., 2018*; *Senum et al., 2022*; *Besse et al., 2019*; *Cornec-Le Gall et al., 2018*; *Porath et al., 2016*). However, not all VEO/EO-ADPKD patients can be explained by monogenic inheritance, suggesting digenic or oligogenic inheritance causes.

Previous data from mouse, *Xenopus,* and zebrafish suggest a crucial role for the RNA-binding protein Bicc1 in the pathogenesis of PKD, although *BICC1* mutations in human PKD have not been previously reported (*Nauta et al., 1993*; *Flaherty et al., 1995*; *Cogswell et al., 2003*; *Maisonneuve et al., 2009*; *Bouvrette et al., 2010*; *Tran et al., 2007*; *Tran et al., 2010*; *Kraus et al., 2012*; *Fu et al., 2010*; *Gamberi et al., 2017*). BICC1 encodes an evolutionarily conserved protein that is characterized by 3 K-homology (KH) and 2 KH-like (KHL) RNA-binding domains at the N-terminus and a SAM domain at the C-terminus, which are separated by a disordered intervening sequence (IVS) (*Dowdle et al., 2022*; *Wessely et al., 2001*; *Wessely and De Robertis, 2000*; *Mahone et al., 1995*; *Rothé et al., 2023*; *Gamberi and Lasko, 2012*). The protein localizes to cytoplasmic foci involved in RNA metabolism and has been shown to regulate the expression of several genes such as *Pkd2, Adcyd6,* and *Pkia* in the kidney (*Tran et al., 2010*; *Piazzon et al., 2012*). We now present data providing a mechanistic model linking BICC1 with the three major cystic proteins. We show that BICC1 physically interacts with the PKD1 (PC1) and the PKD2 (PC2) proteins in human kidney cells. We also demonstrate that *Pkd1* and *Pkd2* modify the cystic phenotype in *Bicc1* mice in a dose-dependent manner and that Bicc1 functionally interacts with Pkd1, Pkd2, and Pkhd1 in the pronephros of *Xenopus* embryos. Finally, this interaction is supported by human patient data where *BICC1* alone or in conjunction with *PKD1* or *PKD2* is involved in VEO-PKD.

## Results

### Interaction of BICC1 with PC1 and PC2

Loss of *Pkd1* has been associated with lower *Bicc1* expression in a murine model (*Lian et al., 2014*). Furthermore, Bicc1 has been shown to regulate *Pkd2* expression in cellular and animal models (*Tran et al., 2010*; *Lemaire et al., 2015*; *Mesner et al., 2014*). However, whether this is due to direct protein-protein interactions between BICC1, PKD1 (PC1), and PKD2 protein (PC2) has not been investigated. In pilot experiments, BICC1 was detected by mass spectrometry in a pulldown assay from cells stably expressing a Polycystin-1 PLAT domain (Polycystin-1, Lipoxygenase, Alpha-Toxin)-YFP fusion (*Xu et al., 2016*). The direct binding between the PC1-PLAT domain and mBicc1 was confirmed using in vitro binding assays, but we also detected binding to the PC1 C-terminus (CT1) (*Figure 1— figure supplement 1a, c, d*).

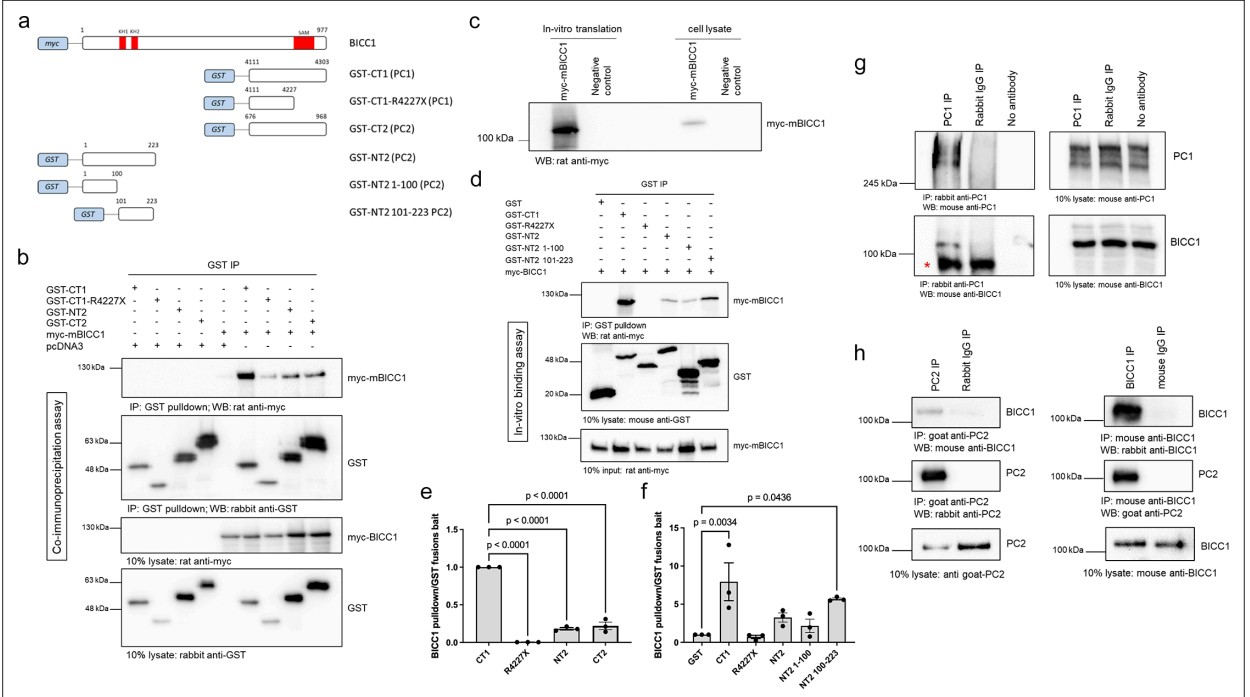

**Figure 1.** mBicc1 forms a complex with Polycystin-1 and Polycystin-2. Full-length and deletion myc-tagged constructs of mBicc1 were co-expressed with either full-length HA-tagged PC1 or PC2 in HEK-293 cells and tested for their ability to interact by co-IP. (**a**) Schematic diagram of the constructs used in this experiment. (**b**) Western blot analysis following co-IP experiments, using GST tagged constructs as bait, identified protein interactions between PC1 or PC2 domains and mBicc1. pcDNA3 was included as a negative control. CT = C-terminus, NT = N-terminus, GST = glutathione S-Transferase. Co-IP experiments (n=3) were quantified in (**e**). (**c**) Western blot showing expression of recombinant myc-tagged mBicc1 generated by in vitro translation or myc-tagged mBicc1 transfected in HEK-293 cells. (**d**) Western blot analysis following in vitro pulldown experiments, using purified GST tagged constructs as bait, identified direct protein interactions between PC1 or PC2 domains and in vitro translated myc-Bicc1. In vitro binding experiments (n=3) were quantified in (**f**). (**g**) Western blot analysis following co-IP experiments, using a rabbit PC1 antibody (2b7) as bait, identified protein interactions between endogenous PC1 and BICC1 in UCL93 cells. A non-immune rabbit IgG antibody or no antibody was included as a negative control; * denotes a non-specific IgG band which is not present in the no antibody control lane. (**h**) Western blot analysis following co-IP experiments, using an anti-BICC1 or anti-PC2 antibody as bait, identified protein interactions between endogenous PC2 and BICC1 in UCL93 cells. Non-immune goat and mouse IgG was included as a negative control.

The online version of this article includes the following source data and figure supplement(s) for figure 1:

**Source data 1.** Original western blots for *Figure 1*, indicating the relevant bands.

**Source data 2.** Original files for western blot displayed in *Figure 1*.

**Figure supplement 1.** In vitro binding assays showing direct binding between Bicc1, PC1-PLAT, and PC1-CT1, but not PC2-CT2.

**Figure supplement 1—source data 1.** Original western blots for *Figure 1—figure supplement 1*, indicating the relevant bands.

**Figure supplement 1—source data 2.** Original files for western blot displayed in *Figure 1—figure supplement 1*.

Utilizing recombinant GST-tagged domains of PC1 and PC2, we demonstrated that mBicc1 binds to both proteins in GST-pulldown assays (*Figure 1a and b*). In the case of PC1, myc-mBicc1 strongly interacted with its C-terminus (GST-CT1), but its interaction was abolished by a PC1-R4227X truncation mutation (GST-CT1-R4227X) (*Figure 1b and c*). In the case of PC2, myc-mBicc1 associated with both recombinant GST N-terminal (GST-NT2) and C-terminal (GST-CT2) fusions. To investigate whether binding was direct or indirect, we performed in vitro binding assays using in vitro translated myc-mBicc1 and recombinant PC1 and PC2 domains. GST-pulldowns confirmed a direct interaction between myc-mBicc1 and GST-CT1 but not GST-CT1-R4227X (*Figure 1d and e*). Similarly, myc-mBicc1 interacted directly with GST-NT2. While binding was stronger with the distal sequence (NT2 aa101-223), both N-terminal fragments contributed to the overall binding to mBicc1 (*Figure 1d and f*). Interestingly, no direct interaction between mBicc1 and GST-CT2 was detected (*Figure 1—figure supplement 1b*), suggesting that the observed in vivo interaction with mBicc1 is indirect. Finally, immunoprecipitation using lysates from human kidney epithelial cells (UCL93) to assay endogenous,

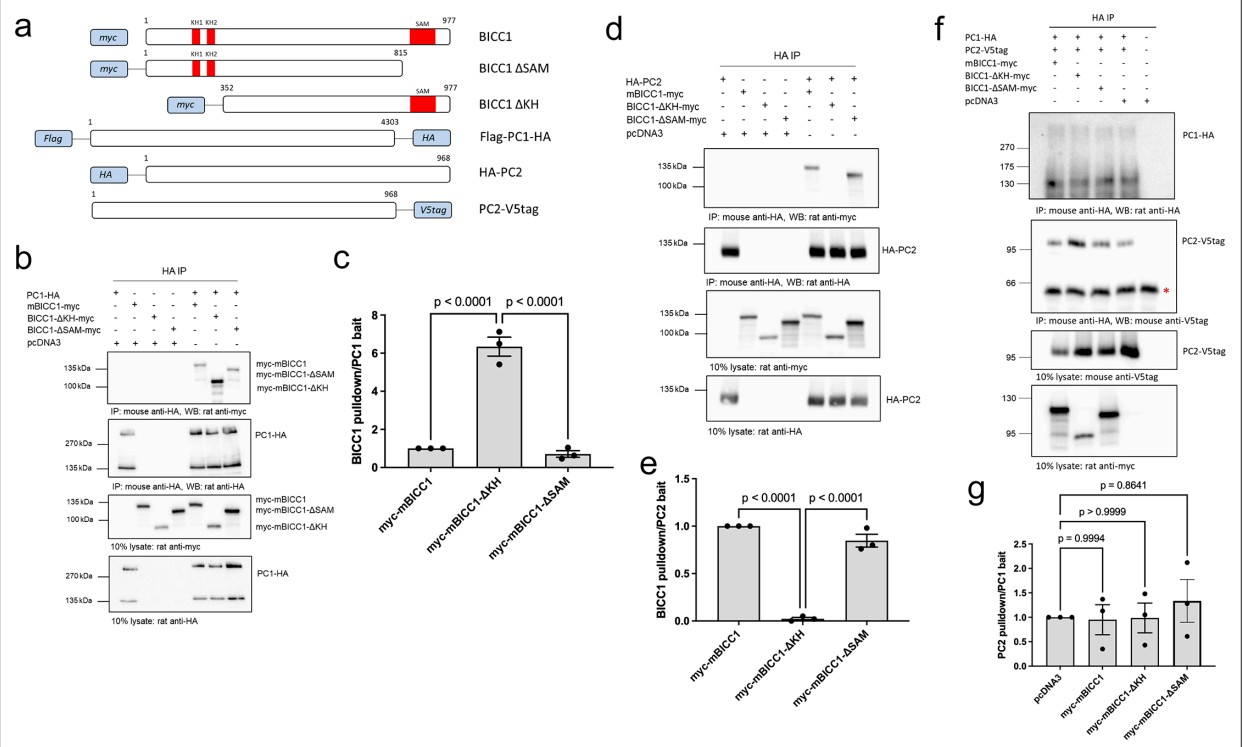

**Figure 2.** Interactions between mBicc1 and Polycystin1/2 require different binding motifs. Full-length and deletion myc-tagged constructs of mBicc1 were co-expressed with either full-length HA-tagged PC1 or PC2 in HEK-293 cells and tested for their ability to interact by co-IP. (**a**) Schematic diagram of the constructs used in this set of experiments with the amino acid positions of full-length mBicc1 or the different deletions indicated. (**b, c**) Western blot analysis following co-IP experiments, using a PC1-HA-tagged construct as bait, identified protein interactions between PC1 and mBicc1 domains. pcDNA3 was included as a negative control (**b**). co-IP experiments (n=3) were quantified in (**c**). (**d, e**) Western blot analysis following co-IP experiments, using a PC2-HA tagged construct as bait, identified protein interactions between PC2 and mBicc1 domains (**d**). pcDNA3 was included as a negative control. Quantification of the co-IP experiments (n=3) is shown in (**e**). (**f, g**) Western blot analysis following co-IP experiments, using a PC1-HA-tagged construct as bait. The interaction between PC1 and PC2 was not altered in the presence of either full-length mBicc1 or its deletion domains. pcDNA3 was included as a negative control. Asterix represents non-specific interaction with mouse IgG. (**f**). co-IP experiments (n=3) were quantified in (**g**). One-way ANOVA comparisons were performed to assess significance, and p values are indicated. Error bars represent standard error of the mean.

The online version of this article includes the following source data for figure 2:

**Source data 1.** Original western blots for *Figure 2*, indicating the relevant bands.

**Source data 2.** Original files for western blot displayed in *Figure 2*.

non-overexpressed proteins showed that PC1, PC2, and BICC1 form protein complexes in vivo (*Figure 1g and h*).

## Different interaction motifs for the binding of mBicc1 to the Polycystins

To define the PC1/PC2 interaction domain(s) in mBicc1, we generated deletion constructs lacking the SAM domain (myc-mBicc1-ΔSAM, aa1-815) or the KH/KHL domains (myc-mBicc1-ΔKH, aa352-977) (*Figure 2a*) and studied them by co-IP. Full-length PC1 co-immunoprecipitated with full-length myc-mBicc1 (*Figure 2b and c*). Deleting the SAM domain did not significantly reduce the association to PC1 (~55%, p=0.79) compared to full-length myc-mBicc1. However, an eightfold stronger interaction was observed between full-length PC1 and myc-mBicc1-ΔKH compared to myc-mBicc1 or myc-mBicc1-ΔSAM. These results suggested that the interaction between PC1 and mBicc1 may involve the SAM but not the KH/KHL domains (nor the first 132 amino acids of mBicc1). Potentially, the N-terminus (aa1-351) could have an inhibitory effect on PC1-mBicc1 association.

Similar experiments were performed to define the mBicc1 interacting domains for PC2 (*Figure 2d and e*). Full-length PC2 (PC2-HA) interacted with full-length myc-mBicc1. Unlike PC1, PC2 interacted with myc-mBicc1-ΔSAM, but not myc-mBicc1-ΔKH, suggesting that PC2 binding is dependent on the

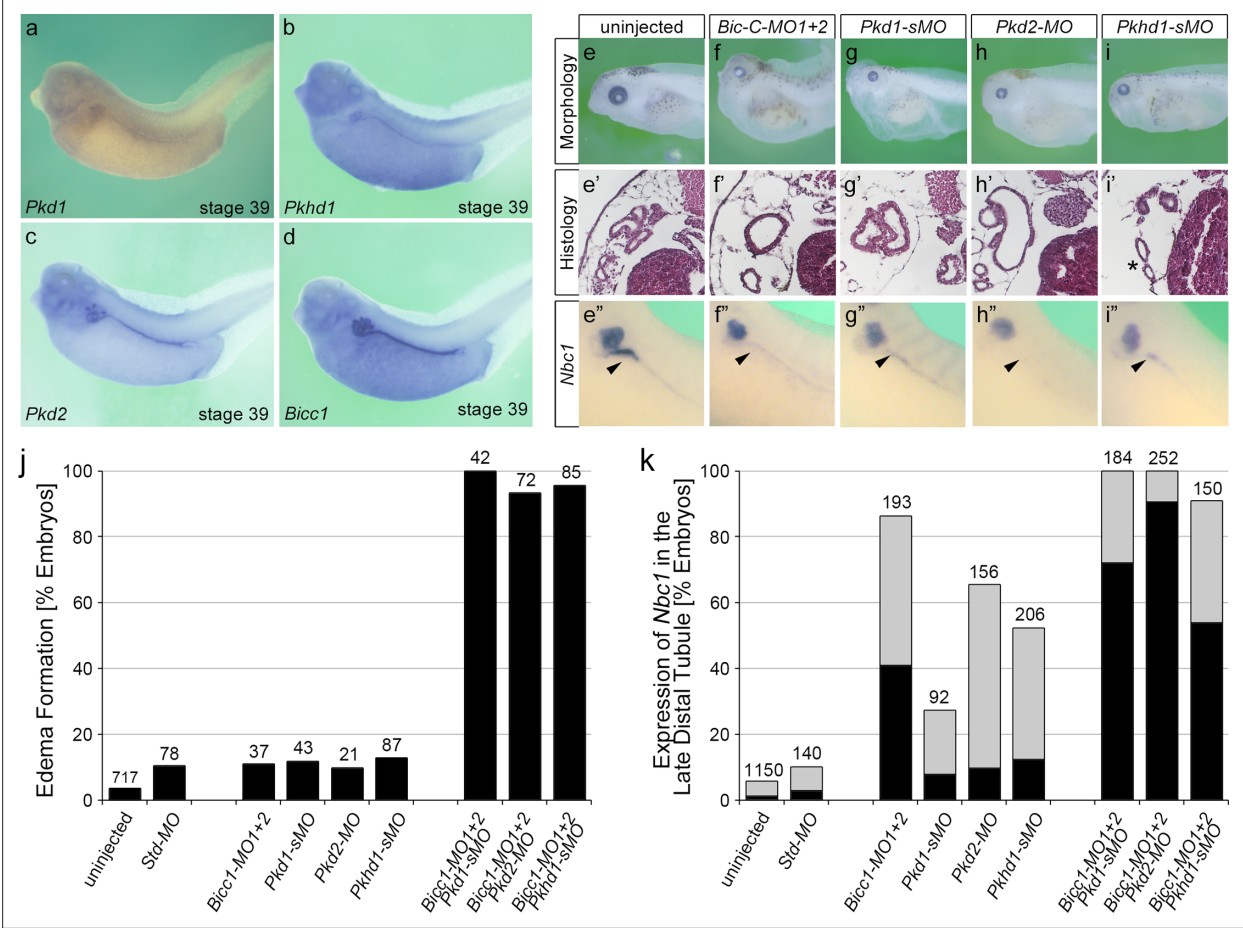

**Figure 3.** Cooperativity of Bicc1 and PKD genes in *Xenopus*. (**a–d**) mRNA expression of *Pkd1*, *Pkhd1*, *Pkd2,* and *Bicc1* in the *Xenopus* pronephros at stage 39. (**e–i"**) Knockdown of Bicc1 (**f–f"**), Pkd1 (**g–g"**), Pkd2 (**h–h"**), and Pkhd1 (**i–i"**) by antisense morpholino oligomers (MOs) results in a PKD phenotype compared to uninjected control *Xenopus* embryos (**e–e"**). The phenotype is characterized by the formation of edema due to kidney dysfunction (**e, f, g, h, i**; stage 43), the development of dilated renal tubules (**e', f', g', h', i'**; stage 43), and the loss of *Nbc1* in the late distal tubule by whole mount in situ hybridizations (arrowheads in **e", f", g", h", i"**; stage 39). (**j, k**) To examine cooperativity, *Xenopus* embryos were injected with suboptimal amounts of the MOs, either alone or in combination, and analyzed for edema formation at stage 43 (**j**) and the expression of *Nbc1* at stage 39 (**k**) with gray bars showing reduced and black bars showing absent *Nbc1* expression in the late distal tubule. Data are the accumulation of multiple independent fertilizations with the number of embryos analyzed indicated above each condition.

The online version of this article includes the following figure supplement(s) for figure 3:

**Figure supplement 1.** Validation of *Xenopus* knockdowns *and* BICC1 knockout.

N-terminal domains (aa1-351) but not the SAM domain or distal C-terminus (aa816-977). Co-expression of mBicc1 deletion constructs lacking the SAM domain (myc-mBicc1-ΔSAM) or the KH domains (myc-mBicc1-ΔKH), however, had no effect on the interaction of PC1 with PC2 in co-immunoprecipitation assays (*Figure 2f and g*), suggesting that these interactions are not mutually exclusive.

## Cooperativity of BICC1 with other PKD genes

Since our biochemical analysis indicated a direct interaction between BICC1, PC1, and PC2, we wondered whether this is biologically relevant. If this were the case, BICC1 should cooperate with other PKD genes, and reducing BICC1 activity in conjunction with reducing either PKD1 or PKD2 activity should still cause a cystic phenotype. We first addressed this question in the *Xenopus* system (*Figure 3*), which is an easily manipulatable model to study PKD. The PKD phenotype in frogs is characterized by dilated kidney tubules, the loss of the expression of the sodium bicarbonate cotransporter 1 (Nbc1) in the distal tubule, and the emergence of body-wide edema as a sign of a malfunctioning kidney (*Tran et al., 2007*; *Tran et al., 2010*; *Xu et al., 2016*; *Naert et al., 2021*). Knockdown

of Bicc1, Pkd1, Pkd2, or the ARPKD protein Pkhd1 caused a PKD phenotype (*Figure 3e–i"* and *Figure 3—figure supplement 1a*). The latter, *Pkhd1,* was included to assay not only ADPKD but also ARPKD, which is generally thought to disturb the same cellular mechanisms. To test whether xBicc1 cooperated with the PKD genes, we then performed combined knockdowns. We titrated each of the four MOs to a concentration that on its own resulted in little phenotypic changes upon injection into *Xenopus* embryos (*Figure 3j, k*, *Figure 3—figure supplement 1b*). However, combining *Bicc1-MO1+2* with *Pkd1-sMO, Pkd2-MO,* or *Pkhd1-sMO* at suboptimal concentrations resulted in the re-emergence of a strong PKD phenotype. While injections with individual MOs developed edema in about 10% of the embryos, co-injections caused edema formation in almost 100% of the embryos (*Figure 3j*, last three columns). A similar result was seen for the expression of *Nbc1* in the late distal tubule, where individual MO injections showed some changes in gene expression, but double MO injections had a highly synergistic effect resulting in a near complete loss of *Nbc1* (*Figure 3k*).

We next investigated whether a similar cooperation between Bicc1 and Pkd1 or Pkd2 can be observed in genetic mouse models. We initially focused on Bicc1 and Pkd2. Both *Bicc1* and *Pkd2* knockout mice develop cystic kidneys as early as E15.5 (*Tran et al., 2010*; *Wu et al., 2000*). As this is the earliest time point cystic kidneys can be observed, crossing those strains did not allow us to assess cooperativity (data not shown). Moreover, like in the case of compound *Pkd1/Pkd2* mutants (*Wu et al., 2002*), kidneys from *Bicc1^{+/-}:Pkd2^{+/-}* not exhibit cysts (data not shown). Thus, we instead used mice carrying the Bicc1 hypomorphic allele *Bpk*, which develop a cystic kidney phenotype postnatally (*Cogswell et al., 2003*; *Nauta et al., 1993*). To assess cooperativity, we removed one copy of *Pkd2* in the *Bpk* mice. Comparing the kidneys of *Bicc1^{Bpk/Bpk}:Pkd2^{+/-}* to those of *Bicc1^{Bpk/Bpk}:Pkd2^{+/+}* at postnatal day P14 revealed that the compound mutant kidneys were larger and more translucent (*Figure 4a*) and the kidney/body weight ratios (KW/BW) were significantly increased (*Figure 4b*). Moreover, analyzing survival, the compound mutants showed a trend towards an earlier demise (*Supplementary file 1a*). We did not detect sex differences in the phenotype (*Figure 4—figure supplement 1c*). Yet, the reduction in *Pkd2* gene dose affected the progression of the disease, but not its onset. Performing the same analysis at postnatal day P4 did not show any differences (*Figure 4c*).

Next, we performed a similar mouse study for *Pkd1* using the *Pkd1^{Fl/Fl}:Pkhd1-Cre* line as previously described (*Williams et al., 2014*) (in the following referred to as *Pkd1^{CD}*). This mouse line eliminates *Pkd1* postnatally in the collecting ducts. Similar to the *Bicc1/Pkd2* scenario, when removing one copy of *Pkd1* in the collecting ducts, the *Bicc1^{Bpk/Bpk}:Pkd1^{+/CD-}* appeared larger when comparing kidneys from littermates (*Figure 4d*) and littermates exhibited statistically significant differences in KW/BW ratio (*Figure 4e*). Yet, the phenotype was rather subtle, and aggregating all the data did not show differences in KW/BW ratios between *Bicc1^{Bpk/Bpk}:Pkd1^{+/+}* and *Bicc1^{Bpk/Bpk}:Pkd1^{+/CD-}* mice (*Figure 4—figure supplement 1d*). Thus, to further corroborate the genetic interaction, we determined the cystic index for proximal tubules and collecting ducts using LTA and DBA staining, respectively. This showed an increase in collecting duct cysts upon removal of one copy of *Pkd1* (*Figure 4g*). Like in the case of *Pkd2*, the phenotype seems to be correlated with cyst expansion and not the onset, as there was no difference at postnatal day P7 (*Figure 4f*) and we did not detect increased mortality in the compound mutants (*Supplementary file 1b*). It is noteworthy that neither the *Bicc1/Pkd2* nor the *Bicc1/Pkd1* compound mutants showed an aggravated kidney function based on blood urea nitrogen (BUN) levels (*Figure 4—figure supplement 1a, b, e*), likely due to the aggressive nature of the *Bicc1^{Bpk/Bpk}* phenotype. Of note, due to the different genetic approaches using a *Pkd2* null allele and a conditional *Pkd1* allele, the outcomes of the two crosses cannot be directly compared. Yet, these in vivo data support our biochemical interaction data and demonstrate that *Bicc1* cooperates with *Pkd1* and *Pkd2*.

Finally, to better understand how Bicc1 would exert such a phenotype, we analyzed the expression of the PKD genes in the *Bicc1^{Bpk/Bpk}* mice. We have previously demonstrated that *Pkd2* levels are reduced in a complete Bicc1 null mice (*Tran et al., 2010*). Performing qRT-PCR of kidneys from wildtype and *Bicc1^{Bpk/Bpk}* at P4 (i.e. before the onset of a strong cystic phenotype) revealed that *Bicc1, Pkd1,* and *Pkd2* were statistically significantly down-regulated (*Figure 4h–j*). The effect on *Pkd2* mRNA was confirmed by protein analysis for PC2 (*Figure 4k*, *Figure 4—figure supplement 1f*).

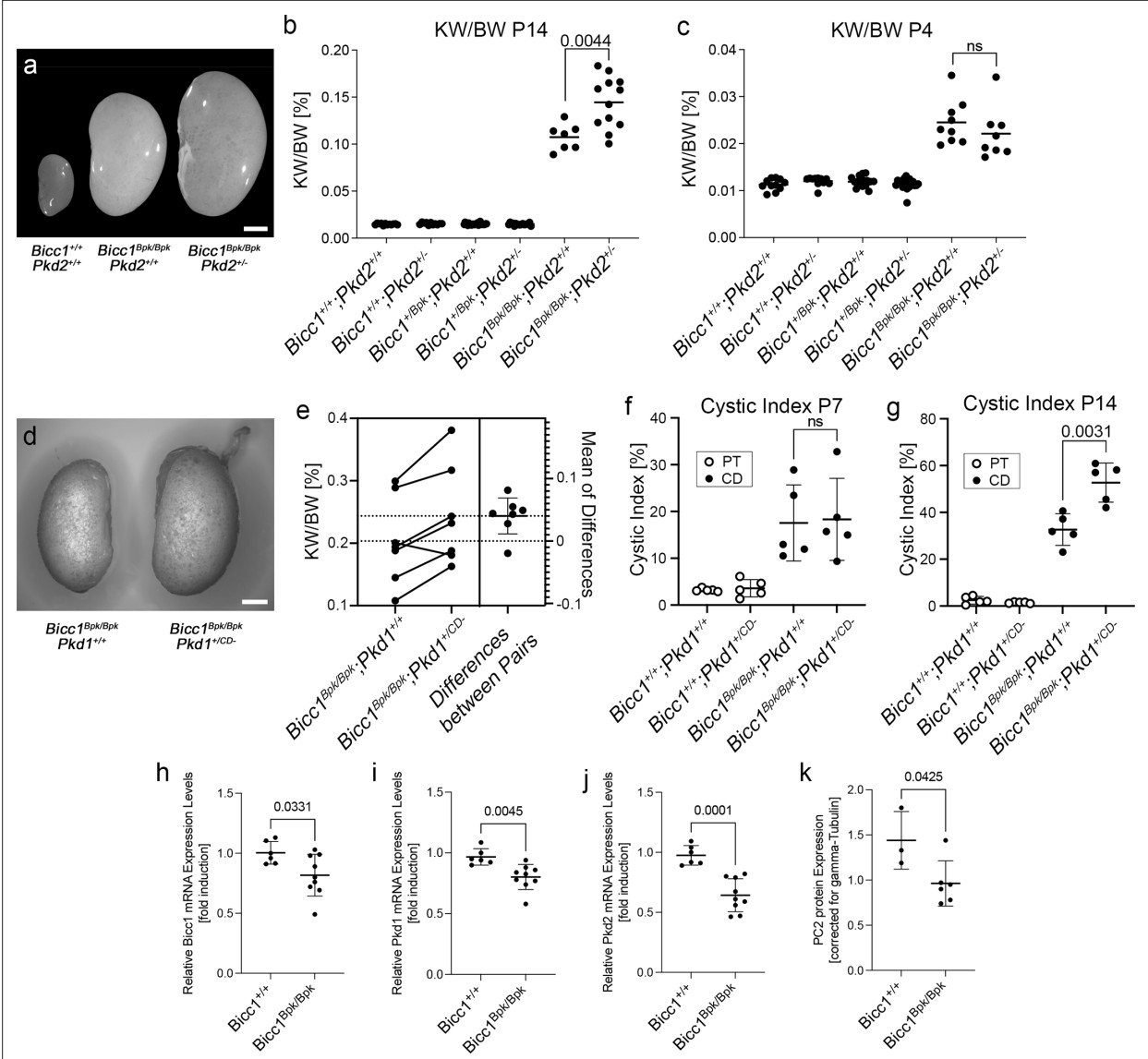

**Figure 4.** Cooperativity of *Bicc1* and *Pkd1* and *Pkd2* in mouse. (**a–c**) Bicc1 and Pkd2 interact genetically. Offspring from *Bicc1;Pkd2* compound mice at postnatal day P4 and P14 are compared by outside kidney morphology at postnatal day P14 (**a**, scale bar is 2 mm), and kidney to body weight ratio (KW/BW) at P14 (**b**) and P4 (**c**). (**d–g**) Bicc1 and Pkd1 interact genetically. *Bicc1;Pkd1* compound mice are compared by outside kidney morphology at P14 showing a kidney from *Bicc1^Bpk/Bpk^:Pkd1^+/+^* and a *Bicc1^Bpk/Bpk^:Pkd1^+/CD-^* littermate (**d**, scale bar is 2 mm, as no wildtype littermate was present in the litter, no wildtype kidney is shown), estimation plot of KW/BW ratio comparing littermates at P14 with a p-value=0.092 (**e**), and cystic index, that is, percent of proximal tubules (PT) and collecting ducts (CD) cysts in respect to the total kidney area at P7 (**f**) and P14 (**g**). Two-sided paired *t*-tests were performed to assess significance, and the p-values are indicated; error bars represent standard deviation. (**h–k**) qRT-PCR analysis for *Bicc1*, *Pkd1*, and *Pkd2* expression (**h–j**) and quantification of the PC2 expression levels by western blot (**k**) in kidneys at P4 before the onset of a strong cystic kidney phenotype. Data were analyzed by *t*-test, and the p-values are indicated. Please note that the y-axes of the different panels are intentionally different to best visualize the changes between the groups analyzed.

The online version of this article includes the following source data and figure supplement(s) for figure 4:

**Figure supplement 1.** Kidney parameters of *Bicc1:Pkd2* and *Bicc1:Pkd1* compound mutants.

**Figure supplement 1—source data 1.** Original western blots for **Figure 4—figure supplement 1**, indicating the relevant bands.

**Figure supplement 1—source data 2.** Original western blots for **Figure 4—figure supplement 1**, indicating the relevant bands.

## *BICC1* variants in patients with early and severe Polycystic Kidney Disease

To evaluate whether these interactions are relevant for human PKD, we analyzed an international cohort of 2914 PKD patients by massive parallel sequencing (MPS) (*Devane et al., 2022*; *Lu et al., 2017*) focusing on VEO-ADPKD patients with the hypothesis that *BICC1* variants may lead to a more severe and earlier PKD phenotype. While variants in *BICC1* are very rare, we could identify two patients with *BICC1* variants harboring an additional *PKD2* or *PKD1* variant in trans, respectively. Moreover, besides the variants reported below, the patients had no other variants in any of the other PKD genes or genes which phenocopy PKD including *PKD1*, *PKD2*, *PKHD1*, *HNF1ß*, *GANAB*, *IFT140*, *DZIP1L*, *CYS1*, *DNAJB11*, *ALG5*, *ALG8*, *ALG9*, *LRP5*, *NEK8*, *OFD1*, or *PMM2*. The first patient was severely and prenatally affected, demonstrating a Potter sequence with huge echogenic kidneys and oligo-/anhydramnios. Autopsy confirmed VEO-ADPKD with absence of ductal plate malformation invariably seen in ARPKD. The fetus carried the *BICC1* variant (c.2462G>A, p.Gly821Glu) inherited from his father, who presented with two small renal cysts in one of his kidneys, and a *PKD2* variant (c.1894T>C, p.Cys632Arg) that arose de novo (*Figure 5a*). Individual in silico predictions (SIFT, Polyphen2, CADD, Eigen-PC, FATHMM, GERP++RS, and EVE), meta scores (REVEL, MetaSVM, and MetaLR) and other protein function predictions (PrimateAI, ESM1b, and ProtVar) indicate that this *PKD2* missense variant is likely pathogenic (*Supplementary file 1c*). Moreover, structural analysis suggests that the hydrophilic substitution may interfere with the Helix S5 pore domain of PKD2 and change its ion channel function (*Figure 5b and c*). Finally, *PKD2* p.Cys632Arg has been previously reported as part of a PKD2 pedigree and implicated as a critical determinant for Polycystin-2 function (*Magistroni et al., 2003*; *Feng et al., 2011*). On the other hand, the *BICC1* p.Gly821Glu variant is located in an intrinsically disordered domain of BICC1 between the KH and the SAM domains (*Figure 6*). To address whether the variant is hypomorphic, we used CRISPR-Cas9-mediated gene editing to generate HEK293T cells lacking BICC1 or harboring the p.Gly821Glu mutation (BICC1-G821E). These cells were analyzed for their impact on the translation of *PKD2*, a well-established target of Bicc1 (*Tran et al., 2010*). As shown in *Figure 5d and e*, PC2 protein levels were strongly reduced in two independent HEK293T BICC1-G821E cells when compared to unedited HEK293T cells. Most notably, the PC2 levels were comparable to the levels found in HEK293T carrying a *BICC1* null allele (HEK293T BICC1-KO) (*Figure 3—figure supplement 1c, d*). Based on these data, we hypothesize that the major disease effect results from the pathogenic *PKD2* variant but is aggravated by the *BICC1* variant.

The second patient presented perinatally with massively enlarged hyperechogenic kidneys, while the parents, both in their thirties, and the remaining family members were reported to be healthy (*Figure 5f–h*). He carried a paternal canonic *BICC1* splicing variant (c.1179+1G>T), which is likely pathogenic as the protein is truncated after exon 10, and a novel heterozygous *PKD1* variant (c.11942C>T, p.Ala3981Val) which has not been previously reported (*Figure 5f*). While the *PKD1* variant appears minor in its amino acid change (i.e., Ala to Val), in silico analyses using individual predictions (SIFT, Polyphen2, CADD and EVE), Meta scores (REVEL) and other protein function predictions (PrimateAI and ESM1b) indicate that the missense variant is likely pathogenic (*Supplementary file 1c*). Structural analyses suggest that although the Ala3981Val variant does not destabilize the Helix structure, its contact with the TOP domain could interfere with domain flexibility and PC1 complex assembly.

## A sibling pair of PKD patients with a homozygous *BICC1* variant

The most insightful finding for a critical role for BICC1 in human PKD was the discovery of a homozygous *BICC1* variant in a consanguineous Arab multiplex pedigree, two siblings, a boy and a girl, diagnosed with VEO-ADPKD (*Figure 6a–e*). The affected female presented at a few months of age with kidney failure and enlarged polycystic kidneys that lacked corticomedullary differentiation. Histology after bilateral nephrectomy showed polycystic kidneys more suggestive of ADPKD than ARPKD without any dysplastic element (*Figure 6c*). Her younger brother exhibited enlarged hyperechogenic polycystic kidneys antenatally by ultrasound (*Figure 6b*). In addition, during early infancy, arterial hypertension and a Dandy–Walker malformation with a low-pressure communicating hydrocephalus were noted (*Figure 6d and e*). By customized MPS, we identified the homozygous missense *BICC1* variant (c.718T>C, p.Ser240Pro) (*Figure 6a*). This variant was absent from gnomAD and fully segregated with the cystic phenotype present in this family. It results in a non-conservative change from the aliphatic, polar-hydrophilic serine to the cyclic, apolar-hydrophobic proline located in the second

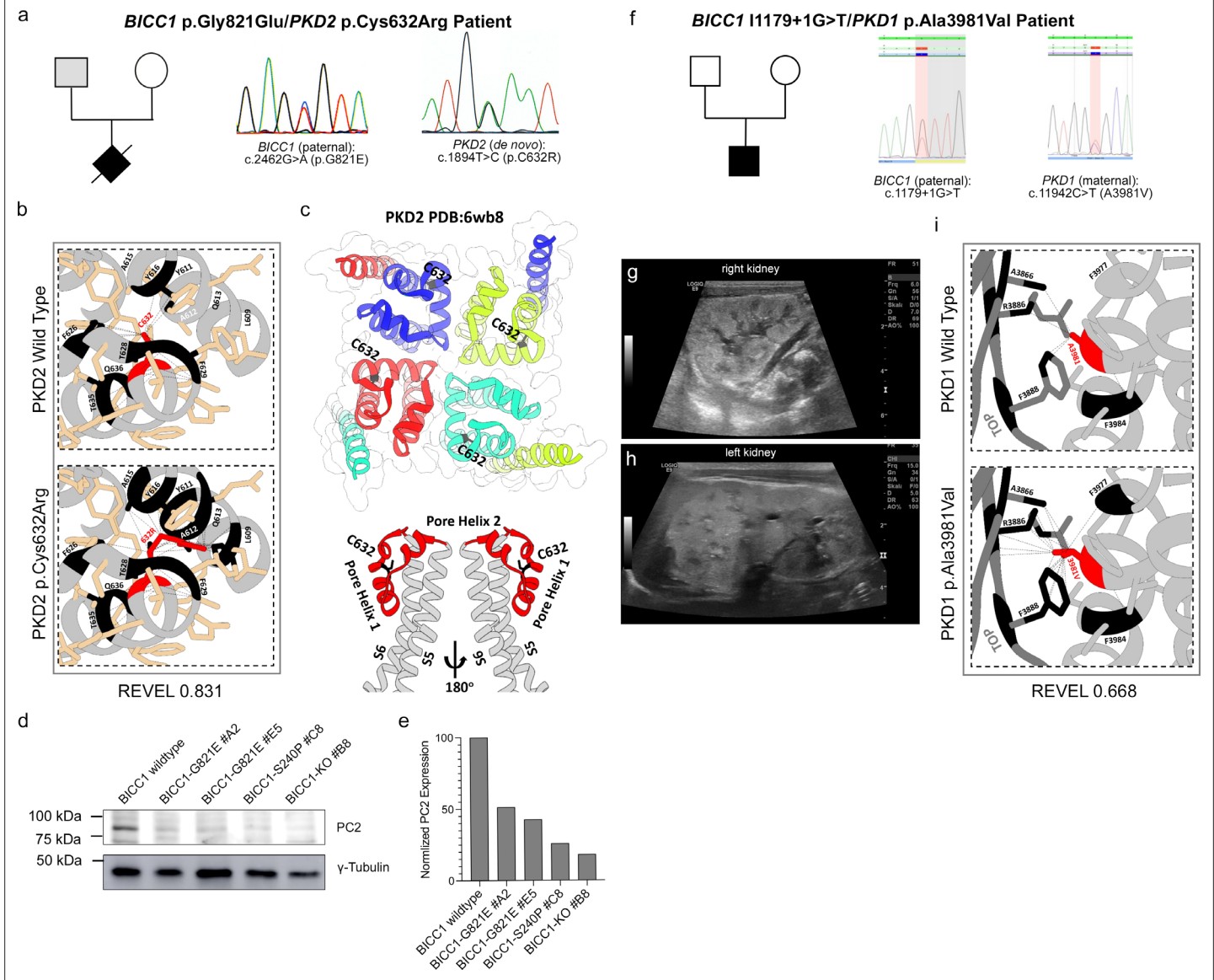

**Figure 5.** Identification of human *BICC1* variants. (**a–c**) *BICC1* p.G821E/*PKD2* p.C632R patient with pedigree and the electropherograms (**a**), the structural analysis of the PKD2 showing the local structure around the cysteine at position 632 (indicated in red) and its putative impact in the variant including the REVEL score (**b**) as well as its location within the global PC2 structure highlighting the potential of the variant impacting the PC2 ion channel function (**c**). (**d, e**) Western blot analysis for PC2 comparing wildtype HEK293T, HEK293T BICC1 p.Gly821Glu (BICC1-G821E), HEK293T BICC1 p.Ser240Pro (BICC1-S240P) and HEK293T BICC1 knockout (BICC1-KO) cells and quantification thereof. γ-Tubulin was used as loading control. (**f–i**) *BICC1* c.1179+1G>T/PKD1 p.Ala3981Val patient with pedigree and the electropherograms (**f**), the ultrasound analysis of the left and right kidneys (**g, h**) and the structural analysis of the PC1 showing the local structure around the alanine at position 3981 (indicated in red) and its putative impact in the variant including the REVEL score (**i**).

The online version of this article includes the following source data for figure 5:

**Source data 1.** Original western blots for *Figure 5*, indicating the relevant bands.

**Source data 2.** Original files for western blot displayed in *Figure 5*.

beta sheet of the first KHL1 domain and very likely disrupts the beta sheet and thus the RNA-binding activity of Bicc1 (*Figure 6f and g* and *Supplementary file 1d*). In the more severely affected younger brother, we also detected an additional heterozygous *PKD2* variant (c.1445T>G, p.Phe482Cys), which results in a non-conservative change from phenylalanine to cysteine (*Supplementary file 1c*). It was previously reported that this PC2 Phe482Cys variant exhibited altered kinetic PC2 channel properties, increased expression in IMCD cells, and a different subcellular distribution when compared to

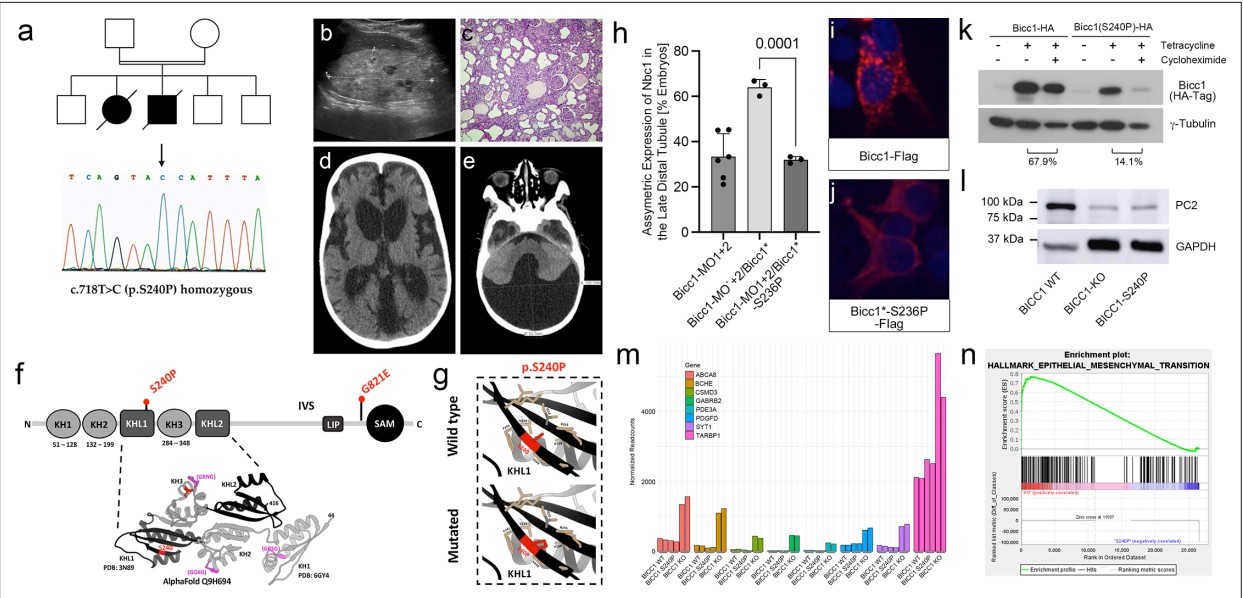

**Figure 6.** The homozygous BICC1 p.Ser240Pro variant is a hypomorphic cystic disease-causing variant. (**a–e**) Consanguineous multiplex pedigree with two siblings affected by VEO-ADPKD identified the homozygous BICC1 missense variant c.718T>C (BICC1 p.Ser240Pro) absent from gnomAD and other internal and public databases. Electropherogram is shown in (**a**). The affected girl presented at a few months of age with renal failure and enlarged polycystic kidneys that lacked corticomedullary differentiation (**c**). Histology after bilateral nephrectomy showed polycystic kidneys more suggestive of ADPKD than ARPKD without any dysplastic element. Her younger brother exhibited enlarged hyperechogenic polycystic kidneys prenatally by ultrasound (**b**). In addition, in his early infancy, arterial hypertension and a Dandy–Walker malformation with a low-pressure communicating hydrocephalus were noted (**d, e**). (**f**) Ribbon diagram and schematic diagram of BICC1 showing the KH, KHL, and SAM domains. The two BICC1 variants identified in this study, BICC1 p.Ser240Pro (S240P) and BICC1 p.Gly821Glu (G821E) are indicated in red. (**g**) Solid boxes correspond to local impacts of p.Ser240Pro (p.S240P) on BICC1 structure, interactions are labeled as dashed lines (pseudobonds). GXXG motifs colored in magenta, representative missense variant residues colored in red and residues adjacent to selected variant (<5 Å) colored in tan. (**h**) Rescue experiments of Xenopus embryos lacking BicC1 by co-injections with the wild type or mutant constructs. Embryos were scored for the re-expression of Nbc1 in the late distal tubule by whole mount in situ hybridizations. Quantification of at least 3 independent experiments is shown. (**i, j**) HEK293T cells were transfected with Flag-tagged constructs of wild type or mutant Bicc1 and the subcellular localization of Bicc1 was visualized (red). Nuclei were counterstained with DAPI (blue). (**k**) Protein stability analysis using tetracycline-inducible HEK293T cells comparing the expression levels of Bicc1 and Bicc1-S240P 24 hours after removal of tetracycline and addition of cycloheximide. γ-Tubulin was used as loading control. The percentage of protein destabilization because of protein synthesis inhibition by cycloheximide is indicated. (**l**) Western Blot analysis of wildtype HEK293T, cells lacking BICC1 (BICC1-KO) and isogenic cells with the BICC1 p.Ser240Pro (BICC1-S240P) variant for PC2 expression. GAPDH was used as loading control. (**m, n**) Bar graph of the mRNA-seq transcriptomic analysis comparing BICC1 wildtype, knockout, and S240P isogenic HEK293T cells showing the eight most significantly upregulated transcripts (based on their Padj levels) in the BICC1 KO cells (**m**). For each gene, the normalized expression levels from each of the 6 samples (2 wildtype, KO, and 240 P each) are shown. (**n**) GSEA plot showing the enrichment of the Hallmark Epithelial_Mesenchymal_Transition data set in the BICC1-KO cells vs. the BICC1-S240P cells.

The online version of this article includes the following source data and figure supplement(s) for figure 6:

**Source data 1.** Original western blots for **Figure 6**, indicating the relevant bands.

**Source data 2.** Original files for western blot displayed in **Figure 6**.

**Figure supplement 1.** Transcriptomic analysis of BICC1 wildtype, BICC1KO, and BICC1-S240P HEK293T cells.

wild-type PC2 (**Dedoussis et al., 2008**). These features suggested altered properties of this PC2 variant, yet its contribution to the case reported here remains untested.

Unfortunately, both siblings passed away, and besides DNA and the phenotypic analysis described above, neither human tissue nor primary patient-derived cells could be collected. Thus, to validate the pathogenicity of this point mutation, we turned to the amphibian model of PKD (**Tran et al., 2007; Tran et al., 2010**). In *Xenopus*, knockdown of Bicc1 using antisense morpholino oligomers (*Bicc1-MO1+2*) causes a PKD phenotype, which can be rescued by co-injection of synthetic mRNA encoding *Bicc1* (**Tran et al., 2007**). To test whether *BICC1* p.Ser240Pro had lost its biological activity, we introduced the same mutation into the *Xenopus* gene where the Ser is located at position 236 of the *Xenopus* gene (in the following referred to as *xBicC1*-S236P). *Xenopus* embryos were injected with *Bicc1-MO1+2* at the two- to four-cell stage followed by a single injection of 2 ng wild type

or *xBicc1\*-S236P* mRNAs at the eight-cell stage. At stage 39 (when kidney development has been completed) embryos were analyzed by whole mount in situ hybridization for the expression of *Nbc1* in the late distal tubule of the pronephric kidney, one of the most reliable readouts for the amphibian PKD phenotype (*Tran et al., 2007*). As shown in *Figure 6h*, wild-type *Bicc1* mRNA restored expression of *Nbc1* on the injected side in 63% of the embryos. However, *xBicc1\*-S236P* did not have any effect, and the embryos were indistinguishable from those injected with the *Bicc1-MO1+2* alone. This suggested that *xBicc1\*-S236P* was functionally impaired. To address this hypothesis, we first assessed the subcellular localization of Bicc1 to foci that are thought to be involved in mRNA processing (*Maisonneuve et al., 2009*; *Tran et al., 2010*; *Rothé et al., 2023*; *Stagner et al., 2009*). Transfection of Flag-tagged Bicc1 (*xBicc1\*-S236P-Flag*) into HEK293T cells reproduced this pattern (*Figure 6i*). Surprisingly, xBicc1\*-S236P-Flag was no longer detected in these cytoplasmic foci but rather homogenously dispersed throughout the cytoplasm (*Figure 6j*). Western blot analysis demonstrated that this was accompanied by a reduction in protein levels (*Figure 6k*). In vitro transcription/translation detected no differences between the proteins, suggesting that the wildtype and xBicc1 S236P-Flag are translated equivalently (data not shown). Yet, in an in vivo pulse-chase experiment, the mBicc1 p.Ser240Pro variant was less stable than its wildtype counterpart (*Figure 6k*). However, whether the reduced protein level was due to an inherent instability of the mutant protein or a consequence of its mislocalization remains to be resolved. Finally, as in the case of BICC1 p.Gly821Glu, we engineered HEK293T cells to harbor the BICC1 p.Ser240Pro variant (BICC1-S240P). Western blot analysis demonstrated a reduction in PC2 levels in the BICC1-S240P cells when compared to unedited cells and that this reduction was comparable to PC2 levels in BICC1-KO cells (*Figures 5d, e and 6l*).

Finally, to determine to what extent the *BICC1* p.Ser240Pro variant differs from a BICC1 loss of function, we performed mRNA sequencing (mRNA-seq) of the genetically engineered HEK293T cells. Differential gene expression analysis identified several genes that were differentially up- or downregulated in the BICC1-S240P and the BICC1-KO cells compared to their unedited counterpart (*Figure 6—figure supplement 1a and e*). Approximately 24% and 18% of the differentially expressed genes were shared between BICC1-S240P or the BICC1-KO cells, respectively (*Figure 6—figure supplement 1*). Yet, a substantial number of genes were specific to either cell line. The BICC1-S240P-enriched/depleted transcripts were generally also enriched/depleted in the BICC1-KO cells but did not reach statistical significance (*Figure 6—figure supplement 1*). Conversely, many of the BICC1-KO enriched transcripts were specifically enriched/depleted in the BICC1-KO cells and not in the BICC1-S240P cells (*Figure 6—figure supplement 1*). This suggested that there are qualitative differences between a null phenotype and the *BICC1* p.Ser240Pro variant, supporting our hypothesis that BICC1 p.Ser240Pro acts as a hypomorph. Indeed, Gene Set Enrichment Analysis (GSEA) using the hallmark gene sets and comparing BICC1-KO and BICC1-S240P cells revealed a statistically significant enrichment for the Hallmark_Epithelial_Mesenchymal_Transition set (*Figure 6n*), a pathway previously implicated in ADPKD (*Kim et al., 2019*; *Formica and Peters, 2020*).

## Discussion

BICC1 has been extensively studied in multiple animal models, which have suggested a critical role for BICC1 in several different developmental processes and in tissue homeostasis (*Dowdle et al., 2022*). This study functionally implicates it to human disease in general and PKD in particular by identifying the homozygous *BICC1* p.Ser240Pro variant, which was sufficient to cause a cystic phenotype in a sibling pair of human PKD patients. It is noteworthy that another study identified heterozygous *BICC1* variants in two patients with mildly cystic dysplastic kidneys (*Kraus et al., 2012*). Yet, both variants were also present in one of the unaffected parents. While such a situation is extremely rare and does not significantly contribute to the mutational load in ADPKD or ARPKD, it demonstrated that loss of BICC1 is sufficient to cause PKD in humans. In addition, variants in *BICC1* and *PKD1 and PKD2* co-segregated in PKD patients from an International Clinical Diagnostic Cohort. While we have not yet shown the impact of each variant when introduced in a compound heterozygous situation, we postulate that PKD alleles in trans and/or de novo exert an aggravating effect and contribute to polycystic kidney disease. A reduced dosage of PKD proteins would severely disturb the homeostasis and network integrity, and by this correlates with disease severity in PKD. ADPKD is quite heterogeneous and – even within the same family – shows quite some phenotypic variation (*Milutinovic et al., 1992*; *Harris and Rossetti, 2010*). It is thought that stochastic inputs, environmental factors, and

genetics influence PKD (*Harris and Rossetti, 2010*). The demonstrated interaction of BICC1, PC1, and PC2 now provides a molecular mechanism that can explain some of the phenotypic variability in these families. Of note, while our mouse studies support cooperation between *Bicc1*, *Pkd1,* and *Pkd2*, genetic proof for Bicc1 acting as a disease modifier, i.e. reduction of Bicc1 activity in a homozygous *Pkd1* or *Pkd2* background in mice remains outstanding.

The second important aspect of this study is that BICC1 emerges as central in the regulation of PKD1/PKD2 activity. Functional studies reported here and previously (*Tran et al., 2010*; *Lemaire et al., 2015*; *Mesner et al., 2014*) demonstrate that Bicc1 regulates the expression of *Pkd1* and *Pkd2*. Moreover, we now show that mBicc1 and PC1/PC2 physically interact and that lowering the expression levels of both proteins is sufficient to cause a PKD phenotype in frogs. Finally, the reduction of the gene dose for *Pkd1* or *Pkd2* in a hypomorphic mouse allele of *Bicc1* results in a more severe cystic kidney phenotype. These results in the kidney are paralleled and augmented in studies of left/right patterning, where Pc2 can activate Bicc1 and where Bicc1 triggers critical aspects in establishing laterality (*Maisonneuve et al., 2009*; *Rothé et al., 2023*; *Minegishi et al., 2021*; *Maerker et al., 2021*). Thus, it is tempting to speculate that BICC1/PC1/PC2 are components of a critical regulatory network in maintaining epithelial homeostasis.

BICC1 has emerged as an important posttranscriptional regulator modifying gene expression through modulating the effects of microRNAs (miRNAs), regulating mRNA polyadenylation and translational repression and activation (*Tran et al., 2010*; *Dowdle et al., 2022*; *Piazzon et al., 2012*; *Wang et al., 2002*; *Chicoine et al., 2007*; *Zhang et al., 2014*; *Zhang et al., 2013*). While *PKD2* is the most appealing target in respect to ADPKD (*Tran et al., 2010*), there are undoubted others (e.g., adenylate cyclase-6) (*Piazzon et al., 2012*) that may be equally critical. Lastly, Bicc1 has been implicated in the regulation of miRNAs such as those of the *miR-17* family (*Tran et al., 2010*). This is of particular interest as a connection between *miR-17* activity and PKD is well-established (*Chu and Friedman, 2008*; *Patel et al., 2013*; *Pandey et al., 2008*; *Pandey et al., 2011*; *Patel et al., 2012*; *Nagalakshmi et al., 2011*; *Yheskel et al., 2019*). Both *Pkd1* and *Pkd2* mRNA are targeted by *miR-17* (*Lakhia et al., 2022*), and an *anti-miR-17* oligonucleotide is being developed as a PKD therapeutic (*Lee et al., 2019*). While we have shown that mBicc1 and *miR-17* targets *Pkd2* mRNA (*Tran et al., 2010*), a similar scenario for *Pkd1* is possible, but not yet shown. Thus, a tempting hypothesis is that the interaction between BICC1, PC1, PC2, and miRNAs - even though not examined in this study – compartmentalizes BICC1's activity where BICC1 is post-transcriptionally inactive when complexed to PC1/PC2 but modulates *PKD1* and *PKD2* translation when unbound. Such a regulatory complex could be responsible for several of the aspects of human ADPKD. In the future, it would be interesting to see how BICC1 and its posttranscriptional targets are integrated and together contribute towards preventing kidney epithelial cells from developing a cystic phenotype.

# Materials and methods

**Key resources table**

| Reagent type (species) or resource | Designation | Source or reference | Identifiers | Additional information |
|---|---|---|---|---|
| Cell line (*Homo sapiens*) | HEK-293 | ETCC and ATTC | | |
| Cell line (*H. sapiens*) | UCL-93 | *Streets et al., 2003* *Parker et al., 2007* | PMID:12819240 PMID:17396115 | |
| Antibody | Anti-Polycystin-1 (7e12, mouse monoclonal) | Santa Cruz Biotechnologies *Ong et al., 1999* | sc-130554, RRID:AB_2163355 PMID:10504485 | Used @ 1:5000 |
| Antibody | Anti-Polycystin-1 (2b7, rabbit polyclonal) | *Newby et al., 2002* | PMID:11901144 | 5 µg used for IP |
| Antibody | Anti-Polycystin-2 (YCC2, rabbit polyclonal) | Kind gift from Dr. S. Somlo | PMID:9568711 | Used @ 1:1000 |

*Continued on next page*

*Continued*

| Reagent type (species) or resource | Designation | Source or reference | Identifiers | Additional information |
|---|---|---|---|---|
| Antibody | Anti-Polycystin-2 (D-3, mouse monoclonal) | Santa Cruz Biotechnologies | sc-28331, RRID:AB_672377 | Used @ 1:1000 |
| Antibody | Anti-Polycystin-2 (G20, goat polyclonal) | Santa Cruz Biotechnologies | sc-10376, RRID:AB_654304 | Used @ 1:1000 |
| Antibody | Anti-myc (JAC6, rat monoclonal) | Bio-Rad | MCA1929, RRID:AB_322203 | Used @ 1:2000 |
| Antibody | Anti-GST (rabbit polyclonal) | Santa Cruz Biotechnologies | sc-459, RRID:AB_631586 | Used @ 1:5000 |
| Antibody | Anti-BICC1 (A-12, mouse monoclonal) | Santa Cruz Biotechnologies | sc-514846, RRID:AB_3717417 | Used @ 1:2000 |
| Antibody | anti-BICC1 (rabbit polyclonal) | Sigma-Aldrich | HPA045212, RRID:AB_10959667 | Used @ 1:2000 |
| Antibody | Anti-$\gamma$-Tubulin (mouse monoclonal) | Sigma-Aldrich | T6557, RRID:AB_477584 | Used @ 1:1000 |
| Antibody | Anti-HA (3F10, rat monoclonal) | Roche | 11867423001, RRID:AB_390918 | Used @ 1:2000 |
| Antibody | Anti-V5-Tag (clone SV5-Pk1, mouse monoclonal) | Bio-Rad | MCA1360, RRID:AB_322378 | Used @ 1:5000 |
| Antibody | Anti-MBP (rabbit polyclonal) | NEB | E8030S, RRID:AB_1559728 | Used @ 1:5000 |
| Antibody | Anti-GST (mouse monoclonal) | Santa Cruz Biotechnologies | sc-138, RRID:AB_627677 | Used @ 1:5000 |
| Antibody | Anti-GAPDH (rabbit monoclonal) | Cell Signaling | 2118, RRID:AB_561053 | Used @ 1:1000 |
| Antibody | Goat Anti-Rabbit IgG(H+L), Mouse/Human ads-HRP | Southern Biotech | 4050-05 | Used @ 1:20,000 |
| Antibody | Mouse IgG1-human ads HRP | Southern Biotech | 1070-05 | Used @ 1:20,000 |
| Antibody | Anti-Rat IgG(H+L) Mouse ads | Southern Biotech | 3050-05 | Used @ 1:20,000 |
| Antibody | Anti-Goat Ig HRP | Dako | P0449 | Used @ 1:20,000 |
| Peptide, recombinant protein | anti-HA mouse conjugated magnetic beads | Thermo Fisher Scientific | 88836 | |
| Peptide, recombinant protein | Protein G Magnetic Beads | Thermo Fisher Scientific | 10003D | |
| Recombinant DNA reagent | myc-mBICC1 | pcDNA3 | Wessely lab PMID:20215348 | |
| Recombinant DNA reagent | myc-mBICC1-$\Delta$KH | pcDNA3 | Ong lab PMID:20168298 PMID:26311459 | |
| Recombinant DNA reagent | myc-mBICC1-$\Delta$SAM | pcDNA3 | Ong lab PMID:20168298 PMID:26311459 | |
| Recombinant DNA reagent | GST-NT2-1-100 | pEBG | Ong lab PMID:20168298 PMID:26311459 | |
| Recombinant DNA reagent | PC1-HA | pcDNA3 | Ong lab PMID:20168298 PMID:26311459 | |
| Recombinant DNA reagent | HA-PC1-R4227X | pcDNA3 | Ong lab PMID:20168298 PMID:26311459 | |

*Continued*

| Reagent type (species) or resource | Designation | Source or reference | Identifiers | Additional information |
|---|---|---|---|---|
| Recombinant DNA reagent | PC2-HA | pcDNA3 | Ong lab PMID:20168298 PMID:26311459 | |
| Recombinant DNA reagent | GST-NT2 101-223 | pEBG | Ong lab PMID:20168298 PMID:26311459 | |
| Recombinant DNA reagent | GST-CT1 | pEBG | Ong lab PMID:20168298 PMID:26311459 | |
| Recombinant DNA reagent | GST-CT1-4227X | pEBG | Ong lab PMID:20168298 PMID:26311459 | |
| Recombinant DNA reagent | GST-NT2 | pEBG | Ong lab PMID:20168298 PMID:26311459 | |
| Recombinant DNA reagent | GST-CT2 | pEBG | Ong lab PMID:20168298 PMID:26311459 | |
| Recombinant DNA reagent | MBP-CT1 | pMAL-c2x | Ong lab PMID:20168298 PMID:26311459 | |
| Recombinant DNA reagent | MBP-CT2 | pMAL-c2x | Ong lab PMID:20168298 PMID:26311459 | |
| Recombinant DNA reagent | MBP-PLAT | pMAL-c2x | Ong lab PMID:20168298 PMID:26311459 | |
| Commercial assay or kit | Omega E.Z.N.A. Plasmid DNA Mini Kit | Omega Bio-Tek | D6942-01 | |

## Cell culture and biochemical studies

The characterization of the interaction between BICC1, PC1, and PC2 as well as the analysis of the human *BICC1* variants were performed using standard approaches detailed in the Appendix 1. The UCL93 kidney epithelial and HEK293T embryonic kidney cells were chosen because of their kidney origin and relevance to the study.

## Animal studies

Mouse and *Xenopus laevis* studies were approved by the Institutional Animal Care and Use Committee at the Cleveland Clinic Foundation (CCF) and LSU Health Sciences Center (LSUHSC), which are the present and the former employer of Dr. Wessely under the following IACUC numbers: 2014-1191 (CCF, mouse study), 2014-1221 (CCF, *Xenopus* study), 2017-1780 (CCF, mouse study), 2017-1802 (CCF, *Xenopus* study), 2019-2307 (CCF, mouse study), 2020-2311 (CCF, *Xenopus* study), 00003071 (CCF, mouse study), 00003105 (CCF, *Xenopus* study) and #2861 (LSUHSC, mouse and *Xenopus* study), #BC0101 (LSUHSC, mouse study) and #2760 (LSUHSC, mouse and *Xenopus* study). Both facilities adhere to the National Institutes of Health Guide for the Care and Use of Laboratory Animals. Experimental design and data interpretation followed the ARRIVE1 reporting guidelines (*Kilkenny et al., 2010*).

## International diagnostic clinical cohort

Research was performed following written informed consent and according to the declaration of Helsinki and oversight was provided by the Medizinische Genetik Mainz. It was performed in

accordance with the German genetic diagnostics act for primarily diagnostic purposes, and consent was given for scientific research and publishing results in a pseudonymized manner. DNA extraction and analysis were performed according to standard procedures (see Appendix 1 for details).

## Statistical analysis

Data are presented as mean values ± SEM. Paired and unpaired two-sided Student's $t$-test or ANOVA were used for statistical analyses with a minimum of $p < 0.05$ indicating statistical significance. Measurements were taken from distinct biological samples. Analyses were carried out using GraphPad Prism 10 (RRID:SCR_000306).

## Acknowledgements

We would like to thank the patients and their families for their cooperation and interest in the study. This work was supported by grants from NIH/NIDDK (R01DK080745) and a philanthropic gift for PKD research at CCF to OW, Kidney Research UK and the PKD Charity UK (PKD_RP_005_20211124), the Sheffield Hospitals Charity and the Sheffield Kidney Research Foundation to AJS and ACMO, the Deutsche Forschungsgemeinschaft (DFG, BE 3910/8-2, BE 3910/9-1, Project-ID 431984000 – Collaborative Research Center SFB 1453), the Federal Ministry of Education and Research (BMBF, 01GM1903I and 01GM1903G) and the European Union's Horizon Europe research and innovation program (grant agreement 101080717, TheRaCil) to CB. DS was supported by a Faculty PhD Scholarship from the University of Sheffield. We thank Drs. S Somlo, P Igarashi, and K Dell for mouse strains, S Feng, and L Chang for technical assistance and R Allen Schweickart for bioinformatical support.

## Additional information

### Competing interests

Eva Decker: Eva Decker is affiliated with Medizinische Genetik Mainz, Limbach Genetics. The author has no other competing interests to declare. Carsten Bergmann: is the Medical and Managing Partner and Director of Medizinische Genetik Mainz, Limbach Genetics. The author has no other competing interests to declare. The other authors declare that no competing interests exist.

### Funding

| Funder | Grant reference number | Author |
| --- | --- | --- |
| National Institute of Diabetes and Digestive and Kidney Diseases | R01DK080745 | Oliver Wessely |
| Kidney Research UK | PKD_RP_005_20211124 | Andrew J Streets Albert CM Ong |
| Deutsche Forschungsgemeinschaft | BE 3910/8-2 | Carsten Bergmann |
| Deutsche Forschungsgemeinschaft | BE 3910/9-1 | Carsten Bergmann |
| Deutsche Forschungsgemeinschaft | SFB 1453/Project-ID 431984000 | Carsten Bergmann |
| Bundesministerium für Forschung, Technologie und Raumfahrt | 01GM1903I | Carsten Bergmann |
| Bundesministerium für Forschung, Technologie und Raumfahrt | 01GM1903G | Carsten Bergmann |
| HORIZON EUROPE European Innovation Council | TheRaCil,Grant Agreement #101080717 | Carsten Bergmann |

| Funder | Grant reference number | Author |
| --- | --- | --- |

The funders had no role in study design, data collection and interpretation, or the decision to submit the work for publication.

## Author contributions

Uyen Tran, Andrew J Streets, Data curation, Formal analysis, Investigation, Writing – review and editing; Devon Smith, Eva Decker, Annemarie Kirschfink, Lahoucine Izem, Jessie M Hassey, Briana Rutland, Manoj K Valluru, Jan Hinrich Bräsen, Elisabeth Ott, Daniel Epting, Tobias Eisenberger, Data curation, Formal analysis, Investigation; Albert CM Ong, Carsten Bergmann, Conceptualization, Supervision, Funding acquisition, Writing – original draft, Project administration, Writing – review and editing; Oliver Wessely, Conceptualization, Data curation, Formal analysis, Supervision, Funding acquisition, Writing – original draft, Project administration, Writing – review and editing

## Author ORCIDs

Uyen Tran ⓘ https://orcid.org/0000-0002-6498-9765
Lahoucine Izem ⓘ https://orcid.org/0000-0001-6498-0579
Daniel Epting ⓘ https://orcid.org/0000-0002-7529-0845
Albert CM Ong ⓘ https://orcid.org/0000-0002-7211-5400
Carsten Bergmann ⓘ http://orcid.org/0000-0002-6061-9759
Oliver Wessely ⓘ https://orcid.org/0000-0001-6440-7975

## Ethics

Research was performed following written informed consent and according to the declaration of Helsinki and oversight was provided by the Medizinische Genetik Mainz. It was performed in accordance with the German genetic diagnostics act for primarily diagnostic purpose, and consent was given for scientific research and publishing results in a pseudonymized manner.

Mouse and Xenopus laevis studies were approved by the Institutional Animal Care and Use Committee at the Cleveland Clinic Foundation (CCF) and LSU Health Sciences Center (LSUHSC) (present and former employer of Dr. Wessely) under the following IACUC numbers: 2014-1191 (CCF, mouse study), 2014-1221 (CCF, Xenopus study), 2017-1780 (CCF, mouse study), 2017-1802 (CCF, Xenopus study), 2019-2307 (CCF, mouse study), 2020-2311 (CCF, Xenopus study), 00003071 (CCF, mouse study), 00003105 (CCF, Xenopus study) and #2861 (LSUHSC, mouse and Xenopus study), #BC0101 (LSUHSC, mouse study) and #2760 (LSUHSC, mouse and Xenopus study). Both facilities adhere to the National Institutes of Health Guide for the Care and Use of Laboratory Animals. Experimental design and data interpretation followed the ARRIVE1 reporting guidelines.

Reviewer #1 (Public review): https://doi.org/10.7554/eLife.106342.3.sa1
Reviewer #2 (Public review): https://doi.org/10.7554/eLife.106342.3.sa2
Reviewer #3 (Public review): https://doi.org/10.7554/eLife.106342.3.sa3
Author response https://doi.org/10.7554/eLife.106342.3.sa4

# Additional files

## Supplementary files

Supplementary file 1. Supplementary tables. (a) Table of the expected vs. observed frequencies in the $Bicc1^{+/Bpk}$:$Pkd2^{+/+}$ x $Bicc1^{+/Bpk}$:$Pkd2^{+/-}$ crosses at P21. (b) Table of the expected vs. observed frequencies in the $Bicc1^{+/Bpk}$:$Pkd1^{+/+}$:$Pkhd1$-Cre+ x $Bicc1^{+/Bpk}$:$Pkd1^{+/fl}$ crosses at P14. (c) Table of the in silico analysis of the PKD1 and PKD2 variants identified in VEO-ADPKD patients. (d) Table of the in silico analysis of the BICC1 p.Ser240Pro (S240P) variant. (e) Table of the gene sets enriched in BICC1-KO vs. BICC1-S240P HEK293T cells.

MDAR checklist

## Data availability

The datasets are presented in the figures and the supplementary information. The mRNA-seq data are deposited into the Gene Expression Omnibus (GEO) database (GSE262417) and are available online. Human exome sequence data are unavailable as they were generated during clinical testing and

individuals were not consented for data sharing. Primary data associated with the study is available at Dryad Digital Repository (https://doi.org/10.5061/dryad.vmcvdnd65).

The following datasets were generated:

| Author(s) | Year | Dataset title | Dataset URL | Database and Identifier |
|---|---|---|---|---|
| Tran U, Izem L, Schweickart RA, Wessely O | 2025 | BICC1 is a genetic modifier for Polycystic Kidney Disease | https://www.ncbi.nlm.nih.gov/geo/query/acc.cgi?acc=GSE262417 | NCBI Gene Expression Omnibus, GSE262417 |
| Wessely O, Tran U, Streets A, Smith D, Decker E, Kirschfink A, Izem L, Hassey J, Rutland B, Valluru M, Bräsen J, Ott E, Epting D, Eisenberger T, Ong A, Bergmann C | 2026 | BICC1 interacts with PKD1 and PKD2 to drive Cystogenesis in ADPKD | https://doi.org/10.5061/dryad.vmcvdnd65 | Dryad Digital Repository, 10.5061/dryad.vmcvdnd65 |

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

# Appendix 1

## Supplementary methods

### Cell culture studies

UCL93 kidney epithelial cells were immortalized from primary cultures of tubular cells isolated from normal human kidneys removed for clinical indications as previously described (*Parker et al., 2007*, *Streets et al., 2003*). Cells were grown in Dulbecco's modified Eagle's medium-Ham's 12 (DMEM-F12, Invitrogen) supplemented with 1% l-glutamine (Invitrogen), 5% NuSerum (Becton Dickinson), and 1% antibiotic/antimycotic solution (Invitrogen) at 33°C/5% $CO_2$. HEK-293 cells were obtained from ATTC (#CRL1573, RRID:CVCL_0045) and were cultured in Dulbecco's modified Eagle's medium-Ham's 12 (DMEM-F12, Invitrogen) supplemented with 1% l-glutamine (Invitrogen), 10% FCS and 1% antibiotic/antimycotic solution (Invitrogen) at 37°C/5% $CO_2$. Cells were transfected using Lipofectamine 3000 (Life Technologies) for 48 hours before the cell assays. Both cell type identities were validated by STR analyses and regularly tested for mycoplasma contamination.

CRISPR/Cas9-mediated knockout and the BICC1 p.Gly821Glu (BICC1-G821E) and BICC1 p.Ser240Pro (BICC1-S240P) knock-in clones in HEK293T cells were generated by Synthego Corporation (Redwood City, CA, USA) with the specifics outlined below. The BICC1 knockout was confirmed by qRT-PCR (*Figure 3—figure supplement 1c*) and, like in the mouse, resulted in a loss of Pkd2 expression that could be rescued by re-expression of mouse Bicc1 (*Figure 3—figure supplement 1d*). In addition, two other genes lost upon elimination of BICC1, *NEFL* and *LAMB3*, were also restored upon re-expression of mouse Bicc1 (*Figure 3—figure supplement 1e and f*). For each engineered cell, two independent clones were generated and analyzed. Data were compared to the mock-transfected parental cell line. Clonal identity was confirmed at regular intervals using the PCR primers indicated below.

### Details on gene editing of HEK293T cells

#### Bicc1 KO

| Cell line | HEK293 |
|---|---|
| Gene name | BICC1 |
| Transcript ID | ENST00000373886.8 |
| Guide RNA sequence | GAGCGAGGAGCGCUUCCGCG |
| Guide RNA cut location | Chr10:58,513,298 |
| Exon targeted | 1 |
| PCR and sequencing primers | FOR primer (5'–3') TGCAGGGGGACGAGCTA<br>REV primer (5'–3') TGGAGCTAAACCGGCCG |
| Sequencing primer | FOR primer (5'–3') TGCAGGGGGACGAGCTA |

### Genotype analysis

1. Clone E1
   Indel: +1
   Description: homozygous KO clone
2. Clone B8

   Indel: –8/+1
   Description: compound heterozygous KO clone

*Continued on next page*

## BICC1 carrying p.Ser240Pro (BICC1-S240P)

| Cell line | HEK293 |
|---|---|
| Gene name | BICC1 |
| Transcript ID | ENST00000373886.8 |
| Guide RNA sequence | UGACAGUAGCACCAUACAUU |
| Guide RNA cut location | Chr 10: 58,789,402 |
| Donor sequence | AACCGGTTCCTGATCCTAATTCCCCCTCTATTCAGCA TATATCACAAAC GTACAATATTTCAGTACCATTTAAA CAGCGTTCACGAATGTATGGTGCT ACTGTCATAGTAC GAGGGTCTCAGAATAACACT |
| PCR and sequencing primers | FOR primer (5'–3') TGCTTTAACTCTCTGCTTTGGA REV primer (5'–3') ACGGGGAAAGATTCTATTGCA |
| Sequencing primer | FOR primer (5'–3') TGCTTTAACTCTCTGCTTTGGA |

### Genotype analysis

1. Clone C8
   Modification: BICC1 p.Ser240Pro (TCA >CCA)
   Description: homozygous KI clone
2. Clone F7

   Modification: BICC1 p.Ser240Pro (TCA >CCA)
   Description: homozygous KI clone

## BICC1 carrying p.Gly821Glu (BICC1-G821E)

| Cell line | HEK293 |
|---|---|
| Gene name | BICC1 |
| Transcript ID | ENST00000373886.8 |
| Guide RNA sequence | GACCGAAAUGGAAUUGGACC |
| Guide RNA cut location | Chr10:58,813,922 |
| Donor sequence | AGCACTTGGGAGGTGGAAGCGAATCTGATAACTGGAGAGACCG AAATGAAA TTGGGCCTGGAAGTCATAGTGAATTTGCAGCTTCTATT GGCAGCCCTAA |
| PCR and sequencing primers | FOR primer (5'–3'): AAAGGCTGTAGGCAGGTTCC REV primer (5'–3'): TCAGAGAGGCCACAGTCAGT |
| Sequencing primer | FOR primer (5'–3'): AAAGGCTGTAGGCAGGTTCC |

### Genotype analysis

1. Clone A2
   Modification: BICC1p.Gly821Glu (GGA >GAA)
   Description: homozygous KI clone
2. Clone E5

   Modification: BICC1 p.Gly821Glu (GGA >GAA)
   Description: homozygous KI clone

### Transcriptome analysis

For mRNA-sequencing, mRNA was extracted using Trizol followed by DNAse treatment. Each cell line/clone was analyzed in triplicates as true technical replicates. Library generation was performed using TruSeq RNA Library Prep Kits (Illumina, San Diego, CA, USA) and sequenced NovaSeq6000 S4 150PE using the services of Psomagen. Primary sequence analysis was performed using Galaxy (*Afgan et al., 2022*). Sequence reads were aligned to the human genome (GRCh38) using STAR (RRID:SCR_004463) in Galaxy (Galaxy Version 2.7.10B+galaxy4, RRID:SCR_006281) with default

parameters. Read counts were obtained using FeatureCounts (Galaxy Version 2.0.3+galaxy2, RRID:SCR_012919) with the default parameters and normalized using DESeq2 (Galaxy Version 2.11.40.8+galaxy0, RRID:SCR_015687) to identify differentially expressed genes (DEGs) and calculate their fold changes (FC), p-values, and false discovery rate (FDR)-adjusted p-values (*Love et al., 2014*). Gene Set Enrichment Analysis (GSEA, RRID:SCR_003199) was used to identify normalized enrichment scores of 50 human hallmark gene sets (*Subramanian et al., 2005*).5 The sequences data are deposited into the Gene Expression Omnibus (GEO, RRID:SCR_005012) database under the accession number GSE262417 and are available online.

## Plasmids

Full-length PC1 and PC2 plasmids used in this article have been previously reported (*Xu et al., 2016*). Polycystin fusion proteins NT2 (PKD2 aa1-223), NT2 1-100 (PKD2 aa1-100), NT2 101-223 (PKD2 aa101-223), CT2 (PKD2 aa680-968), PLAT (PKD1 aa3118-3223), and CT1 (PKD1 aa4107-4303) were subcloned into pGEX-6P-1, pEBG, or pMAL-c2X vectors to express N-terminal bacterial, mammalian GST-fusion proteins, or MBP-fusion proteins respectively (*Xu et al., 2016*; *Giamarchi et al., 2010*). myc-mBicc1-ΔSAM (BICC1 aa1-815) and myc-mBicc1-ΔKH (BICC1 aa352-977) truncations were generated by PCR cloning from full-length myc-mBicc1 plasmid. All plasmids were verified by Sanger sequencing. Of note, we have adapted a spelling of Bicc1, where BICC1 is the human homologue, mBicc1 is the mouse homologue, and xBicc1 the *Xenopus* one.

## Antibodies

Primary antibodies used in this study were mouse anti-BICC1 mAb (clone A12, Santa Cruz Biotechnologies, sc-514846), rabbit anti-BICC1 (Sigma-Aldrich, HPA045212, RRID:AB_10959667), mouse anti-PC1 mAb (clone 7e12, Santa Cruz Biotechnologies, sc-130554, RRID:AB_2163355) (*Ong et al., 1999*), rabbit anti-PC1 (clone 2b7) (*Newby et al., 2002*), goat anti-PC2 (sc-10376, Santa Cruz), rabbit anti-PC2 Ab (YCC2, a kind gift from Dr. S. Somlo or Santa Cruz Biotech, SC-28331, RRID:AB_672377), rat anti-HA (clone 3F10, Roche, 11867423001, RRID:AB_390918), mouse anti-GST mAb (Santa Cruz Biotechnologies, sc-138, RRID:AB_627677), rat anti-Myc (clone JAC6, Bio-Rad, MCA1929, RRID:AB_322203), mouse anti-V5-Tag mAb (clone SV5-Pk1, Biorad, MCA1360, RRID:AB_322378), rabbit anti-GAPDH mAb (clone 14C10, Cell Signaling, 2118, RRID:AB_561053) and mouse anti-γ-Tubulin mAb (clone GTU-88, Sigma-Aldrich, T6557, RRID:AB_477584). All primary antibodies were used at 1:1000 unless otherwise stated. Secondary antibodies used in this study include goat anti-mouse IgG (1030-05, Southern Biotech), goat anti-rabbit IgG (4050-01, Southern Biotech), goat anti-rat IgG (3050-01, Southern Biotech), and rabbit anti-goat IgG (P0449, Dako). All secondary antibodies were used at 1:10,000, unless otherwise stated in the results section.

## Protein biochemistry

Cells were lysed by extraction at 4°C using the IP lysis Buffer (25 mM NaCl, 150 mM EDTA, 1 mM 0.5% NP40, 1% Triton X-100, pH 7.0) supplemented with a protease inhibitor cocktail (Roche). Immunoblotting and immunoprecipitation were performed as previously described (*Newby et al., 2002*). Biorad ChemiDocXRS+ and Image Lab 5.1 software were used for visualization and quantification of proteins of interest. All quantification was carried out on non-saturated bands as determined by the software from three independent experiments.

## Recombinant protein preparation

Plasmids were transformed into the *Escherichia coli* strain BL21-RIPL, and recombinant protein expression was induced at 37°C for 3 hours with 0.5 mM IPTG. MBP-tagged, GST fusion, and His-tagged proteins were purified with Amylose, Glutathione-Sepharose, or Nickel columns, respectively, as previously described (*Giamarchi et al., 2010*).

## Preparation of in vitro translated Bicc1

Myc-tagged *mBicc1* was in vitro transcribed and translated with a reticulocyte lysate system TnT SP6 (Promega, USA). Briefly, the plasmid DNA (1 μg) and 50 μl of the reaction mixture were incubated for 90 minutes at 30°C. Expression of myc-mBicc1$_{IVT}$ was determined by western blotting.

## GST pull-down assays

1–2 µg of the bacterial GST fusion protein and 10 µl myc-mBicc1 $_{IVT}$ were incubated in 300 µl binding buffer (1×TBST with 0.2% Tween20) for 1 hour at room temperature (RT) with gentle rotation. 40 µl of 50% Glutathione Sepharose 4B beads (GE Healthcare) were then added and the mixture was incubated with rotation for an additional hour. The beads were sedimented by centrifugation at 6000 rpm for 2 minutes and washed up to six times with 1 ml volumes of ice-cold PBS. Bound proteins were eluted either using 25 µl of elution buffer or by boiling for 5–10 minutes in reducing sample buffer.

## *Xenopus* embryo manipulations

*Xenopus laevis* (RRID:NCBITaxon_8355) embryos obtained by in vitro fertilization were maintained in 0.1× modified Barth medium (*Sive et al., 2000*) and staged according to *Nieuwkoop and Faber, 1994*. *Xenopus* experiments, we performed injections using at least three independent clutches per experimental group. Final numbers of animals/experimental group varied as survival was clutch-dependent, and animals that did not gastrulate properly or were severely malformed were excluded from subsequent analysis. Microinjections were performed on randomly selecting cleaving embryos at the two- to four-cell stage for a given antisense MO/MO combination. Data analysis was performed in a blinded fashion, and groups were only revealed post data acquisition. The sequences of the antisense morpholino oligomers (GeneTools, LLC) used in this study were 5'-GGG ACA AAG ATG CTC ATT TTA ACA G-3' (*BicC-MO1*) (*Tran et al., 2007*), 5'-GCC ACT ATC TCT TCA ATC ATC TCC G-3' (*BicC-MO2*) (*Tran et al., 2007*), 5'-TCC TTA TGG TCC GAG TTA CCT TGG G-3' (*Pkd1-sMO*) (*Xu et al., 2016*; *Zhang et al., 2011*), 5'- GGT TTG ATT CTG CTG GGA TTC ATC G-3' (*Pkd2-MO*) (*Tran et al., 2010*), and 5'- TAT TGT GTT CTA TTC TTA CCT TTC T-3' (*Pkhd1-sMO*). For complete knockdown, a total of 3.2 pMol of *Std-MO, Pkd1-sMO, Pkd2-MO, Pkhd1-sMO,* or a mixture of 3.2 pMol *Bic-C-MO1* and 3.2 pMol *Bic-C-MO2* (*Bic-C-MO1+2*) was injected radially at the two- to four-cell stage into *Xenopus* embryos. Note that *Xenopus laevis* is allotetraploid, and while we normally target both the L and S allele with one MO, in the case of Bicc1, it requires two. For suboptimal knockdowns, 0.8 pMol of the *Bic-C-MO1*, *Bic-C-MO2*, *Pkd1-sMO,* or *Pkd2-MO* and 0.4 pMol *Pkhd1-sMO* were used. Knockdown of Pkd1 and Pkhd1 was performed using MOs targeting 3' splice donor sites (*Pkd1-sMO* and *Pkhd1-sMO*). Microinjection assays and RT-PCR demonstrated that both splice MOs are functional and prevent proper splicing of the two genes (*Figure 3—figure supplement 1a* and Supplementary Figure S12 in *Xu et al., 2016*). Suboptimal concentrations were determined by injecting serially diluted MOs and determining the concentration-dependent induction of the edema phenotype (*Figure 3—figure supplement 1b*). Of note, the combinatorial knockdown approach is based on a sensitized biological readout, but not on reducing expression levels to a fixed amount such as, for example, 50%.

For synthetic mRNA, *pCS2-xBicC*\* (*Tran et al., 2007*) and its derivatives carrying the corresponding point mutations (generated by Quikchange II Mutagenesis kit from Stratagene) were linearized with *Not*I and transcribed with SP6 RNA polymerase using the mMessage mMachine (Ambion). Rescue experiments, whole mount in situ hybridizations, and histology were performed as previously described (*Tran et al., 2007*). To generate antisense probes, the plasmids were linearized and transcribed as follows: pSK-Bicc1 (*Wessely and De Robertis, 2000*) – NotI/T7, pCMV-SPORT6-Nbc1 (*Zhou and Vize, 2004*) – SalI/T7, pGEM-T-Easy-Pkd1 – NcoI/Sp6, pCRII-TOPO-Pkd2 (*Tran et al., 2010*) – NotI/Sp6, pGEM-T-Easy-Pkhd1 – NcoI/Sp6.

## Mouse studies

For the mouse studies, the sample sizes for the experimental groups were not determined a priori using a power analysis as we did not know the effect sizes for the phenotypes under investigation. Thus, we collected multiple litters until the number of the mutant phenotypes was statistically significantly different from the controls and the number of animals in the experimental groups of interest exceeded 10. Genotyping was performed after collecting the biological data; thus, the investigator was blinded during the data acquisition phase. No outliers were removed unless mice were moribund before sacrifice. In addition, we parsed the data based on sex as a biological variable but did not detect any differences. The Pkd2/Bicc1 mouse crosses were performed using two mouse strains, one carrying the hypomorphic Bicc1 allele Bpk (*Nauta et al., 1993*) and one of a Pkd2

null allele (*Wu et al., 1998*). As the two mice strains were of different genetic background, that is, BALB/c (RRID:MGI:2683685) and C57BL/6 (RRID:IMSR_JAX:000664), we utilized a breeding scheme minimizing the influence of the genetic background. Bicc1$^{+/Bpk}$ and Pkd2$^{+/-}$mice were crossed to generate Bicc1$^{+/Bpk}$:Pkd2$^{+/-}$compound heterozygotes as F1 generation. These mice were then intercrossed to generate the experimental animals in the F2 generation. Mice were genotyped by PCR and analyzed at postnatal day P4, P14, and P21. Kidneys were examined as previously described (*Tran et al., 2010*) for kidney function using BUN (QuantiChrom Urea Assay Kit, BioAssay Systems), morphometric parameters (body and kidney weight) as well as histology and immunofluorescence analyses (i.e., Lotus tetragonolobus agglutinin [LTA] and Dolichos biflorus agglutinin [DBA] to determine cyst origin). Cystic index was calculated as percent of the kidney occupied by proximal (LTA-positive) or collecting duct (DBA-positive) cysts.

The *Pkd1/Bicc1* mouse crosses were performed using the same *Bicc1* hypomorphic allele *Bpk*, which was transferred into the C57BL/6 background by backcrossing for more than 10 generations. The *Bpk* allele displayed the same cystic kidney phenotype in this background as the one described for BALB/c (*Akbari et al., 2022*). These mice were intercrossed to the *Pkd1$^{fl/fl}$;Pkhd1-Cre* mice (a kind gift from Drs. Somlo and Igarashi), an allele we refer to as *Pkd1$^{CD-}$* in this study. Kidneys were analyzed at postnatal day P7 and P14 for kidney function, morphometric parameters, histology, and immunofluorescence, as described for the *Bicc1/Pkd2* mutants.

Of note, the choice of the mouse strains was based on the availability of mice at the time of the experiments and not due to scientific reasons. As we had not finished backcrossing the Bicc1-Bpk strain from Balb/c into C57BL/6, it would have been scientifically unsound to assume genetic homogeneity and cross them with the Pkd2 mutant mice in an uncontrollable fashion. Thus, the interaction between Bicc1 and Pkd2 was performed by generating breeders (Bicc1$^{+/Bpk}$:Pkd2$^{+/+}$ and Bicc1$^{+/Bpk}$:Pkd2$^{+/-}$) in the F1 generation and the experimental animals in the F2 generation. Yet, when we started exploring the interaction between Bicc1 and Pkd1, all three mouse strains (Bicc1$^{+/Bpk}$, Pkd1$^{fl/fl}$ and Pkhd1-Cre) were available in the C57BL/6 strain and the Bicc1$^{+/Bpk}$ had been backcrossed into C57BL/6 more than 10 generations. Thus, the Bicc1/Pkd1 study was performed using traditional breeding schemes.

## International diagnostic clinical cohort

Next Generation Sequencing (NGS) technologies and comprehensive bioinformatic analyses utilized in this project are described in detail elsewhere (*Devane et al., 2022*; *Lu et al., 2017*). In brief, we performed different NGS-based approaches utilizing a customized sequence capture library with curated target regions – currently comprising more than 650 genes described and associated with cystic kidney disease or allied disorders – as well as corresponding flanking intronic sequence according to the manufacturer's recommendations. The panel design is enriched by targets in non-coding regions for described variants listed in well-accepted databases like HGMD or ClinVar (RRID:SCR_006169) and optimized for low-performance and disease-critical regions (e.g., *PKD1*). DNA samples were enriched using sequence capture, multiplexed, and in most cases sequenced using Illumina sequencing-by-synthesis technology with an average coverage of more than 300×. Raw data were processed following bioinformatics best practices. Mapping and coverage statistics were generated from the mapping output files using standard bioinformatics tools (e.g., Picard). Statistical analysis was conducted on our internal database currently comprising >20,000 datasets. The total of this data pool is summarized over samples analyzed by NGS-based customized panel testing or whole exome sequencing (WES) analysis. Customized panel setups have been regularly updated. Sub-cohorts of patients were categorized based on clinical, ultrasound, and/or histologic data. Control cohorts were selected by ruling out any involvement of kidney-related symptoms. This approach yielded high and reproducible coverage enabling copy number variation (CNV) analysis. The performance of the wet-lab and bioinformatic processes is validated and controlled according to national and international guidelines (*Chicoine et al., 2007*; *Zhang et al., 2014*) reaching high sensitivity for SNV, Indels, and CNVs using well-established reference samples, as well as a large cohort of positive controls, especially for CNVs (*Matthijs et al., 2016*; *Rehm et al., 2013*). For interpretation of identified variants, we established a bioinformatic algorithm automatically calculating ACMG classification based on existing and updated guidelines (*Ellard et al., 2020*; *Richards et al., 2015*) and was conducted according to specific standardized internal

procedures. Bioinformatically called variants were classified according to ACMG/AMP and ACGS guidelines in respect to current literature and database entries (internal and external mutation and frequency databases, public clinical and functional studies) as well as family history and – if available – segregation results. Variant prioritization was based on this classification and on the frequency of the respective variants in public databases. Variants e.g., in the genes *PKD1, PKD2,* and *BICC1* were filtered and prioritized for very rare variants in external (gnomAD) and internal databases in our cohort of patients with PKD, classified as pathogenic, likely pathogenic, or VUS, not present in the overall control cohort of all patients in our database and/or patients not affected by PKD or a similar phenotype. Sequence variants of interest were verified by Sanger sequencing, if NGS results failed internal validation guidelines.

For statistical analyses of our patient data, we screened our entire internal database. In a control sub-cohort rigorously screened against any clinical involvement of kidney symptoms (>10,700 patients), neither a *BICC1* variant (class III–V) in combination with a *PKD1* or *PKD2* variant nor a relevant monoallelic *BICC1* variant could be identified using the workflow used for variant prioritization described above. We also repeated both queries on cohorts of patients clinically presenting as glomerular disease/focal segmental glomerular sclerosis (FSGS) or atypical hemolytic uremic syndrome (aHUS) with 957 and 1889 cases and datasets, respectively. Again, we did not detect a single patient with any of the variants described in the article.

## In silico studies

The 3D structure of BICC1 (UniProt: Q9H694), PKD1 (UniProt: P98161) and PKD2 (UniProt: Q13563) was downloaded from PDB (6GY4, 4RQN, Bicaudal-C ortholog GLD-3 '3N89', 6A70 and 6WB8), modeled by AlphaFold (RRID:SCR_025454) and the PHYRE2 automated protein homology modeling server (*Nakel et al., 2010*, *Rothé et al., 2018*, *Kelley et al., 2015*, *Jumper et al., 2021*). Because no experimentally mutant BICC1 structures have been determined, we generated mutant structures by individually introducing the missense mutations in silico; missense mutations were then computationally modeled in UCSF Chimera 1.14 (*Pettersen et al., 2004*) by first swapping amino acids using optimal configurations in the Dunbrack rotamer library (*Shapovalov and Dunbrack, 2011*) and by taking into account the most probable rotameric conformation of the mutant residue. All kinds of direct interactions, that is, polar and nonpolar, favorable and unfavorable, including clashes, were analyzed using the contacts command in UCSF Chimera 1.14 (*Pettersen et al., 2004*). The evolutionary conservation score of each amino acid of BICC1 in its conserved domains (KH, KHL, and SAM domains) was determined using the ConSurf algorithm, based on the phylogenetic relationships between sequence homologues (*Ashkenazy et al., 2016*). To determine the effects of the mutations in flexible conformations of the protein, we used DynaMut, a consensus predictor of protein stability based on the vibrational entropy changes predicted by an elastic network contact model (ENCoM) (*Rodrigues et al., 2018*). Pathogenicity of the variants was predicted using Ensembl Variant Effect Predictor (VEP, RRID:SCR_007931) (*McLaren et al., 2016*) to calculate a REVEL score (*Ioannidis et al., 2016*) and the structural impact of missense variants analyzed using VarSite (*Laskowski et al., 2020*). The pathogenicity score of BICC1, PKD1, and PKD2 variants was also determined using different predictors with the scores collated from Argus dbNSFP and ProtVar (*Schröter et al., 2023*; *Liu et al., 2020*).

