## [Editor Report · eLife Assessment]

This study presents **valuable** findings regarding the basic molecular pathways leading to the cystogenesis of Autosomal Dominant Polycystic Kidney Disease, suggesting BICC1 functions as both a minor causative gene for PKD and a modifier of PKD severity. **Solid** data were supplied to show the functional and structural interactions between BICC1, PC1 and PC2, respectively. The characterization of such interactions remains to be developed further, which renders the specific relevance of these findings for the etiology of relevant diseases unclear.

---

## [Referee Report · Reviewer #1 (Public review)]

In this manuscript, Tran et al. investigate the interaction between BICC1 and ADPKD genes in renal cystogenesis. Using biochemical approaches, they reveal a physical association between Bicc1 and PC1 or PC2 and identify the motifs in each protein required for binding. Through genetic analyses, they demonstrate that Bicc1 inactivation synergizes with Pkd1 or Pkd2 inactivation to exacerbate PKD-associated phenotypes in Xenopus embryos and potentially in mouse models. Furthermore, by analyzing a large cohort of PKD patients, the authors identify compound BICC1 variants alongside PKD1 or PKD2 variants in trans, as well as homozygous BICC1 variants in patients with early-onset and severe disease presentation. They also show that these BICC1 variants repress PC2 expression in cultured cells.

Overall, the concept that BICC1 variants modify PKD severity is plausible, the data are robust, and the conclusions are largely supported.

Comments on revision:

My comments have been mostly addressed.

---

## [Referee Report · Reviewer #2 (Public review)]

Tran and colleagues report evidence supporting the expected yet undemonstrated interaction between the Pkd1 and Pkd2 gene products Pc1 and Pc2 and the Bicc1 protein in vitro, in mice, and collaterally, in Xenopus and HEK293T cells. The authors go on to convincingly identify two large and non-overlapping regions of the Bicc1 protein important for each interaction and to perform gene dosage experiments in mice that suggest that Bicc1 loss of function may compound with Pkd1 and Pkd2 decreased function, resulting in PKD-like renal phenotypes of different severity. These results led to examining a cohort of very early onset PKD patients to find three instances of co-existing mutations in PKD1 (or PKD2) and BICC1. Finally, preliminary transcriptomics of edited lines gave variable and subtle differences that align with the theme that Bicc1 may contribute to the PKD defects, yet are mechanistically inconclusive.

These results are potentially interesting, despite the limitation, also recognized by the authors, that BICC1 mutations seem exceedingly rare in PKD patients and may not "significantly contribute to the mutational load in ADPKD or ARPKD". The manuscript has several intrinsic limitations that must be addressed.

The manuscript contains factual errors, imprecisions, and language ambiguities. This has the effect of making this reviewer wonder how thorough the research reported and analyses have been.

Comments on revision:

My comments have been addressed.

---

## [Referee Report · Reviewer #3 (Public review)]

Summary:

This study investigates the role of BICC1 in the regulation of PKD1 and PKD2 and its impact on cytogenesis in ADPKD. By utilizing co-IP and functional assays, the authors demonstrate physical, functional, and regulatory interactions between these three proteins.

Strengths:

(1) The scientific principles and methodology adopted in this study are excellent, logical, and reveal important insights into the molecular basis of cystogenesis.

(2) The functional studies in animal models provide tantalizing data that may lead to a further understanding and may consequently lead to the ultimate goal of finding a molecular therapy for this incurable condition.

(3) In describing the patients from the Arab cohort, the authors have provided excellent human data for further investigation in large ADPKD cohorts. Even though there was no patient material available, such as HUREC, the authors have studied the effects of BICC1 mutations and demonstrated its functional importance in a Xenopus model.

Weaknesses:

This is a well-conducted study and could have been even more impactful if primary patient material was available to the authors. A further study in HUREC cells investigating the critical regulatory role of BICC1 and potential interaction with mir-17 may yet lead to a modifiable therapeutic target.

Conclusion:

The authors achieve their aims. The results reliably demonstrate the physical and functional interaction between BICC1 and PKD1/PKD2 genes and their products.

The impact is hopefully going to be manifold:

(1) Progressing the understanding of the regulation of the expression of PKD1/PKD2 genes.

Comments on revision:

My comments have been addressed and sorted.

---

## [Author Response]

The following is the authors’ response to the original reviews.

**Reviewer #1 (Public review):**
(1) The authors devote significant effort to characterizing the physical interaction between Bicc1 and Pkd2. However, the study does not examine or discuss how this interaction relates to Bicc1's well-established role in posttranscriptional regulation of Pkd2 mRNA stability and translation efficiency.

The reviewer is correct that the present study has not addressed the downstream consequences of uthis interaction considering that Bicc1 is a posttranscriptional regulator of Pkd2 (and potentially Pkd1). We think that the complex of Bicc1/Pkd1/Pkd2 retains Bicc1 in the cytoplasm and thus restrict its activity in participating in posttranscriptional regulation (see Author response image 1). We, however, do not yet have data to support this and thus have not included this model in the manuscript. Yet, we have updated the discussion of the manuscript to further elaborate on the potential mechanism of the Bicc1/Pkd1/Pkd2 complex.

We have updated the discussion to include a discussion on the potential consequences on posttranscriptional regulation by Bicc1.

**Author response image 1. sa4fig1:** Model of BICC1, PC1 and PC2 self-regulation. In this model Bicc1 acts as a positive regulator of PKD gene expression. In the presence of ‘sufficient’ amounts of PC1/PC2 complex, it is tethered to the complex and remains biologically inactive (Fig. 1A). However, once the levels of the PC1/PC2 complex are reduced, Bicc1 is now present in the cytoplasm to promote expression of the PKD proteins, thereby raising their levels (Fig. 4B), which then in turn will ‘shutdown’ Bicc1 activity by again tethering it to the plasma membrane.

(2) Bicc1 inactivation appears to downregulate Pkd1 expression, yet it remains unclear whether Bicc1 regulates Pkd1 through direct interaction or by antagonizing miR-17, as observed in Pkd2 regulation. This should be further examined or discussed.

This is a very interesting comment. Vishal Patel published that PKD1 is regulated by a mir-17 binding site in its 3’UTR (PMID: 35965273). We, however, have not evaluated whether BICC1 participates in this regulation. A definitive answer would require utilization of the mice described in above reference, which is beyond the scope of this manuscript. We, however, have revised the discussion to elaborate on this potential mechanism.

We have updated the discussion to include a statement on the potential direct regulation of Pkd1 mRNA by Bicc1.

(3) The evidence supporting Bicc1 and ADPKD gene cooperativity, particularly with Pkd1, in mouse models is not entirely convincing, likely due to substantial variability and the aggressive nature of Bpk/Bpk mice. Increasing the number of animals or using a milder Bicc1 strain, such as jcpk heterozygotes, could help substantiate the genetic interaction.

We have initially performed the analysis using our Bicc1 complete knockout, we previously reported on (PMID 20215348) focusing on compound heterozygotes. Yet, similar to the Pkd1/Pkd2 compound heterozygotes (PMID 12140187) no cyst development was observed when we sacrificed the mice as late as P21. Our strain is similar to the above mentioned jcpk, which is characterized by a short, abnormal transcript thought to result in a null allele (PMID: 12682776). We thank the reviewer for pointing us to the reference showing the heterozygous mice exhibit glomerular cysts in the adults (PMID: 7723240). This suggestion is an interesting idea we will investigate. In general, we agree with the reviewer that a better understanding of the contribution of Bicc1 to the adult PKD phenotype will be critical. To this end, we are currently generating a floxed allele of Bicc1 that will allow us to address the cooperativity in the adult kidney, when e.g. crossed to the Pkd1^RC/RC^ mice. Yet, these experiments are beyond the timeframe for this revision.

No changes were made in the revised manuscript.

**Reviewer #2 (Public review):**
(1) These results are potentially interesting, despite the limitation, also recognized by the authors, that BICC1 mutations seem exceedingly rare in PKD patients and may not "significantly contribute to the mutational load in ADPKD or ARPKD". The manuscript has several intrinsic limitations that must be addressed.

As mentioned above, the study was designed to explore whether there is an interaction between BICC1 and the PKD1/PKD2 and whether this interaction is functionally important. How this translates into the clinical relevance will require additional studies (and we have addressed this in the discussion of the manuscript).

(2) The manuscript contains factual errors, imprecisions, and language ambiguities. This has the effect of making this reviewer wonder how thorough the research reported and analyses have been.

We respectfully disagree with the reviewer on the latter interpretation. The study was performed with rigor. We have carefully assessed the critiques raised by the reviewer. As presented below, most of the criticisms raised by the reviewer have been easily addressed in the revised version of the manuscript. Yet, none of the critiques seems to directly impact the overall interpretation of the data.

**Reviewer #1 (Recommendations for the authors):**
(1) The manuscript requires further editing. For example, figure panels and legends are mismatched in Figure 1

We have corrected the labeling of Figure 1.

(2) Y-axis units and values are inconsistent in Figures 4b-4g, Supplementary Figures S2e and S2f are not referenced in the text, genotypes are missing in Supplementary Figure S3f, and numerous typographical errors are present.

In respect to the y-axis in Figure 4b-g, the scale is different for each of them, but that is intentional as one would lose the differences if they were all scaled identically. But we have now mentioned this in the figure legend to make the reader aware of it. In respect to the Supplemental Figure S2e,f, we included the panels in the description of the mutant BICC1 lines, but unfortunately forgot to reference them. This has now been done.

We have updated the labeling of the Y-axis for the cystic indices adding “[%]” as the unit and updated the figure legend of Figure 4. We have included the genotypes in Supplementary Figure S3f. The Supplementary Figure S2e,f is now mentioned in the supplemental material (page 9, 2^nd^ paragraph).

**Reviewer #2 (Recommendations for the authors):**
(1) Previous data from mouse, Xenopus, and zebrafish suggest a crucial role for the RNAbinding protein Bicc1 in the pathogenesis of PKD, although BICC1 mutations in human PKD have not been previously reported." The cited sources (and others that were not cited) link Bicc1 mutations to renal cysts, similar to a report by Kraus (PMID: 21922595) that the authors cite later. However, a more direct link to PKD was reported by Lian and colleagues using whole Pkd1 mice (PMID: 20219263) and by Gamberi and colleagues using Pkd1 kidneys and human microarrays (PMID: 28406902). Although relevant, neither is cited here, and only the former is cited later in the manuscript.

Thanks for pointing this out. We have added these three citations.

We have added these three citations (PMID: 21922595, PMID: 20219263 and PMID: 28406902) in the indicated sentence.

(2) In Figure 1B, the lanes do not seem to correspond among panels, particularly evident in the panel with myc-mBicc1. Hence, it is difficult to agree with the presented conclusions.

We have corrected the labeling of the lanes in Figure 1b.

(3) In the Figure 1 legend: "(g) Western blot analysis following co-IP experiments, using an anti-mouse Bicc1 or anti-goat PC2 antibody as bait, identified protein interactions between endogenous PC2 and BICC1 in UCL93 cells. Non-immune goat and mouse IgG were included as a negative control." There is no mention of panel H, although this reviewer can imagine what the authors meant. The capitalization differs in the figure and legend. More troublingly, in panel G, a non-defined star indicates a strong band present in both immune and non-immune control.

We have corrected the figure legend of Figure 1 and clarified the non-specific band in the figure legend.

(4) In Figure 4, the authors do not show the matched control for the Bicc1 Pkd1 interaction in panel d, nor do they show a scale bar in either (a) or (d). Thus, the phenotypic severity cannot be properly assessed.

Thanks for pointing out the missing scale bars, which have now been added. In respect to the two kidneys shown in Figure 4d, the two kidneys shown are from littermates to illustrate the kidney size in agreement with the cumulative data shown in Figure 4e. Unfortunately, this litter did not have a wildtype control. As the data analysis in Figure 4e is based on littermates, mixing and matching kidneys of different litters does not seem appropriate. Thus, we have omitted showing a wildtype control in this panel. However, the size of the wildtype kidney can be seen in Figure 4a.

We have added the scale bar to both panels and have updated the figure legend to emphasize that the kidneys shown are from littermates and that no wildtype littermate was present in this litter.

(5) "Surprisingly, an 8-fold stronger interaction was observed between full-length PC1 and myc-mBicc1-ΔKH compared to mycmBicc1 or myc-mBicc1-ΔSAM." Assuming all the controls for protein folding and expression levels have been carried out and not shown/mentioned, this sentence seems to contradict the previous statement that Bicc1deltaSAM reduced the interaction with PC1 by 55%. Because the full length and SAM deletion have different interaction strengths, the latter sentence makes no sense.

The reduction in the levels of myc-mBicc1-ΔSAM compared to wildtype mycmBicc1 in respect to PC1 binding was not significant. We have clarified this in the text.

We have corrected the sentence and modified the Figure accordingly.

(6) Imprecise statements make a reader wonder how to interpret the data: "More than three independent experiments were analyzed." Stating the sample size or including it in the figure would save space and improve confidence in the data presented.

We have stated the exact number of animals per conditions above each of the bars.

(7) "Next, we performed a similar mouse study for Pkd1 by reducing the gene dose of Pkd1 postnatally in the collecting ducts using a Pkhd1-Cre as previously described40" What did the authors mean?

The reference was included to cite the mouse strain, but realized that it can be mis-interpreted that the exact experiments has been performed previously. We have clarified this in the text.

We have reworded the sentence to avoid misinterpretation.

(8) The authors examined the additive effects of knocking down Bicc1, Pkd1, and Pkd2 with morpholinos in Xenopus and, genetically, in mice. While the Bicc1[+/-] Pkd1 or 2[+/-] double heterozygote mice did not show phenotypes, the authors report that the Bicc1[-/-] Pkd1 or 2 [+/-] did instead show enlarged kidneys. What is the phenotype of a Bicc1[+/-] Pkd1 or 2 [-/-]? What we learn from the author's findings among the PKD population suggests that the latter situation would be potentially translationally relevant.

The mouse experiments were designed to address a cooperativity between Bicc1 and either Pkd1 or Pkd2 and whether removal of one copy of Pkd1 or Pkd2 would further worsen the Bicc1 cystic kidney phenotype. Thus, the parental crosses were chosen to maximize the number of animals obtained for these genotypes. Unfortunately, these crosses did not yield the genotypes requested by the reviewer. To address the contribution of Bicc1 towards the PKD population, we will need to perform a different cross, where we eliminate Pkd1 or Pkd2 in a floxed background of Bicc1 postnatally in adult mice. While we are gearing up to perform such an experiment, this is timewise beyond the scope of the manuscript. In addition, please note that we have addressed the question about the translation towards the PKD population already in the discussion of the original submission (page 13/14, last/first paragraph).

No changes have been made to the revised version of the manuscript.

(9) How do the authors interpret the milder effects of the Bicc1[-/-] Pkd1[+/-] compared to Bicc1[-/-] Pkd2[+/-] relative to the respective protein-protein interactions?

The milder effects are due to the nature of the crosses. While the Pkd2 mutant is a germline mutation, the Pkd1 mutant is a conditional allele eliminating Pkd1 only in the collecting ducts of the kidney. As such, we spare other nephron segments such as the proximal tubules, which also significantly contribute to the cyst load. As such these mouse data support the interaction between Pkd1 and Pkd2 with Bicc1, but do not allow us to directly compare the outcomes. While this was mentioned in the previous version of the manuscript, we have expanded on this in the revised version of the manuscript.

We have expanded the results section in the revised version of the manuscript highlighting that the two different approaches cannot be directly compared.

(10) How do the authors interpret that the strong Bicc1[Bpk] Pkd1 or Pkd2 double heterozygote mice did not have defects and "kidneys from Bicc1+/-:Pkd2+/- did not exhibit cysts (data not shown)", when the VEO PKD patients and - although not a genetic reduction - also the morpholino-treated Xenopus did?

VEO PKD patients are characterized by a loss of function of PKD1 or PKD2 and – as we propose in this manuscript - that BICC1 further aggravates the phenotype. Yet, we do not address either in the mouse or Xenopus experiments whether BICC1 is a genetic modifier. We are simply addressing whether the two genes show a genetic interaction. In the mouse studies, we eliminate one copy of Pkd1 or Pkd2 in the background of a hypomorphic allele of Bicc1. Similarly, in the Xenopus experiments, we employ suboptimal doses of the morpholino oligomers, i.e., concentrations that did not yield a phenotypic change and then asked whether removing both together show cooperativity. It is important to state that this is based on a biological readout and not defined based on the amount of protein. While we have described this already in the original manuscript (page 7, first paragraph), we have amended our description of the Xenopus experiment to make this even clearer.

Finally, we agree with the reviewer that if we were to address whether Bicc1 is a modifier of the PKD phenotype in mouse, we would need to reduce Bicc1 function in a Pkd1 or Pkd2 mutants. Yet, we have recognized this already in the initial version of the manuscript in the discussion (page 14, first paragraph).

We have expanded the results section when discussing the suboptimal amounts of the morpholino oligos (Page 6, 1^st^ paragraph).

(11) Unclear: "While variants in BICC1 are very rare, we could identify two patients with BICC1 variants harboring an additional PKD2 or PKD1 variant in trans, respectively." Shortly after, the authors state in apparent contradiction that "the patients had no other variants in any of other PKD genes or genes which phenocopy PKD including PKD1, PKD2, PKHD1, HNF1s, GANAB, IFT140, DZIP1L, CYS1, DNAJB11, ALG5, ALG8, ALG9, LRP5, NEK8, OFD1, or PMM2."

The reviewer is correct. This should have been phrased differently. We have now added “Besides the variants reported below” to clarify this more adequately.

The sentence was changed to start with “Besides the variants reported below, […].”

(12) "The demonstrated interaction of BICC1, PC1, and PC2 now provides a molecular mechanism that can explain some of the phenotypic variability in these families." How do the authors reconcile this statement with their reported ultra-rare occurrence of the BICC1 mutations?

As mentioned in the manuscript and also in response to the other two reviewers, Bicc1 has been shown to regulate Pkd2 gene expression in mice and frogs via an interaction with the miR-17 family of microRNAs. Moreover, the miR-17 family has been demonstrated to be critical in PKD (PMID: 30760828, PMID: 35965273, PMID: 31515477, PMID: 30760828). In fact, both other reviewers have pointed out that we should stress this more since Bicc1 is part of this regulatory pathway. Future experiments are needed to address whether Bicc1 contributes to the variability in ADPKD onset/severity. Yet, this is beyond the scope of this study.

Based on the comments of the two other reviewers we have further addressed the Bicc1/miR-17 interaction.

(13) The manuscript should use correct genetic conventions of italicization and capitalization. This is an issue affecting the entire manuscript. Some exemplary instances are listed below.(a) "We also demonstrate that Pkd1 and Pkd2 modifies the cystic phenotype in Bicc1 mice in a dose-dependent manner and that Bicc1 functionally interacts with Pkd1, Pkd2 and Pkhd1 in the pronephros of Xenopus embryos." Genes? Proteins?

The data presented in this section show that a hypomorphic allele of Bicc1 in mouse and a knockdown in Xenopus yields this. As both affect the proteins, the spelling should reflect the proteins.

No changes have been made in the revised manuscript.

(b) The sentence seems to use both the human and mouse genetic capitalization, although it refers to experiments in the mouse system “to define the Bicc1 interacting domains for PC2 (Fig. 2d,e). Full-length PC2 (PC2-HA) interacted with full-length myc-mBICC1.”

We agree with the review that stating the species of the molecules used is critical, we have adapted a spelling of Bicc1, where BICC1 is the human homologue, mBicc1 is the mouse homologue and xBicc1 the Xenopus one.

We have highlighted the species spelling in the methods section and labeled the species accordingly throughout the manuscript and figures.

(14) “Together these data supported our biochemical interaction data and demonstrated that BICC1 cooperated with PKD1 and PKD2.” Are the authors implying that these results in mice will translate to the human protein?

We agree that we have not formally shown that the same applies to the human proteins. Thus, we have changed the spelling accordingly.

We have revised the capitalization of the proteins.

(15) The text is often unclear, terse, or inconsistent.(a) “These results suggested that the interaction between PC1 and Bicc1 involves the SAM but not the KH/KHL domains (or the first 132 amino acids of Bicc1). It also suggests that the N-terminus could have an inhibitory effect on PC1-BICC1 association.” How do the authors define the N-terminus? The first 132 aa? KH/KHL domains?

This was illustrated in the original Figure 2A. The DKH constructs lack the first 351 amino acids.

To make this more evident, we have specified this in the text as well.

(b) Similarly, the authors state below, "Unlike PC1, PC2 interacted with mycmBICC1ΔSAM, but not myc-mBICC1-ΔKH suggesting that PC2 binding is dependent on the N-terminal domains but not the SAM domain." It is unclear if the authors refer to the KH/KHL domains or others. Whatever the reference to the N-terminal region, it should also be consistent with the section above.

This is now specified in the text.

(c) Unclear: "We have previously demonstrated that Pkd2 levels are reduced in a complete Bicc1 null mice,22 performing qRT-PCR of P4 kidneys (i.e. before the onset of a strong cystic phenotype), revealed that Bicc1, Pkd1 and Pkd2 were statistically significantly down9 regulated (Fig. 4h-j)".

We have changed the text to clarify this.

(d) “Utilizing recombinant GST domains of PC1 and PC2, we demonstrated that BICC1 binds to both proteins in GST-pulldown assays (Fig. 1a, b)." GST-tagged domains? Fusions?

We have changed the text to clarify this.

(e) "To study the interaction between BICC1, PKD1 and PKD2 we combined biochemical approaches, knockout studies in mice and Xenopus, genetic engineered human kidney cells" > genetically engineered.

We have changed the text to clarify this.

(f) Capitalization (e.g., see Figure S3, ref. the Bpk allele) and annotation (e.g., Gly821Glu and G821E) are inconsistent.

We have homogenized the labeling of the capitalization and annotations throughout the manuscript.

(g) What do the authors mean by "homozygous evolutionarily well-conserved missense variant"?

We have changed this is the revised version of the manuscript.

**Reviewer #3 (Public review/Recommendations to the authors):**
(1) A further study in HUREC cells investigating the critical regulatory role of BICC1 and potential interaction with mir-17 may yet lead to a modifiable therapeutic target.(2) This study should ideally include experiments in HUREC material obtained from patients/families with BICC1 mutations and studying its effects on the PKD1/2 complex in primary cell lines.

This is an excellent suggestion. We agree with the reviewer that it would have been interesting to analyze HUREC material from the affected patients. Unfortunately, besides DNA and the phenotypic analysis described in the manuscript neither human tissue nor primary patient-derived cells collected once the two patients with the BICC1 p.Ser240Pro variant passed away.

No changes to the revised manuscript have been made to address this point.

(3) Please remove repeated words in the following sentence in paragraph 2 of the introduction: "BICC1 encodes an evolutionarily conserved protein that is characterized by 3 K-homology (KH) and 2 KH-like (KHL) RNA-binding domains at the N-terminus and a SAM domain at the C-terminus, which are separated by a by a disordered intervening sequence (IVS).23-28".

This has been changed.